# Aluminum corrosion–passivation regulation prolongs aqueous batteries life

Binghang Liu [1,2,3], Tianshi Lv[1,2,3], Anxing Zhou[1,2,3], Xiangzhen Zhu[1,2,3], Zejing Lin[1,2], Ting Lin [1] & Liumin Suo [1,2,3] ✉

Aluminum current collectors are widely used in nonaqueous batteries owing to their cost-effectiveness, lightweightness, and ease of fabrication. However, they are excluded from aqueous batteries due to their severe corrosion in aqueous solutions. Here, we propose hydrolyzation-type anodic additives to form a robust passivation layer to suppress corrosion. These additives dramatically lower the corrosion current density of aluminum by nearly three orders of magnitude to ~$10^{-6}$ A cm$^{-2}$. In addition, realizing that electrochemical corrosion accompanies anode prelithiation, we propose a prototype of self-prolonging aqueous Li-ion batteries (Al $\|$LiMn$_2$O$_4$ $\|$TiO$_2$), whose capacity retention rises from 49.5% to 70.1% after 200 cycles. A sacrificial aluminum electrode where electrochemical corrosion is utilized is introduced as an electron supplement to prolong the cycling life of aqueous batteries. Our work addresses the short-life issue of aqueous batteries resulting from the corrosion of the current collector and lithium loss from side reactions.

Aqueous Li-ion batteries (ALIBs) show promise as large-scale energy storage technology due to their nonflammability and environmental friendliness[1–6]. However, their cycle life is inadequate for the demand in energy storage due to irreversible parasitic side reactions from their cathode and anode sides.

The cathode side expectedly features cycling fading in the form of irreversible oxidation corrosion of metal current collectors in aqueous electrolytes[4,6–9]. Thus, cost-effective, manufacture-friendly, and light-weight Al foils are excluded as current collectors in ALIBs because of the harsh aqueous environment attacks on the Al$_2$O$_3$ passivation film and continuous corrosion of Al, which damage the electronic matrix and result in contact loss and high resistance of electrodes[10]. To address these issues, scholars have used durable metal current collectors, including titanium[11], nickel[12], stainless steel[3,9], and carbon foil[13–17], to reduce the effects of corrosion on ALIBs. However, almost all of these candidates fail as substitutes to Al current collectors, which can simultaneously satisfy the lightweight, conductivity, mechanical strength, and cost-effectiveness requirements.

Different from nonaqueous electrolytes that passivate Al through the addition of effective salts, such as LiPF$_6$, LiBF$_4$, LiDFOB, or optimization of solvents with a low dielectric constant[10,18–28], the passivation of Al metal foil in aqueous solutions is considerably more challenging because of the incompatibility of the salts and water. On the one hand, solvent water possesses a relatively high dielectric constant (78.4 F m$^{-1}$ at 25 °C), which promotes the diffusion of corrosion products and accelerates Al corrosion[18,19]; on the other hand, the formation of solid electrolyte interphase (SEI) in aqueous electrolytes highly depends on organic salts, such as bis(trifluoromethane sulfonyl) imide (LiTFSI), and lithium triflouromethanesulfonate (LiOTF)[3], whose anions are aggressive against Al regardless of the use of nonaqueous or aqueous electrolytes[18–20,26–29]. Therefore, addressing the corrosion of Al by aqueous LiTFSI electrolyte poses an important challenge in the field of aqueous batteries. Although the corrosion of Al current collectors can be suppressed through the increase in salt concentration[4,6] since high concentration and high viscosity decrease the dissolving capability and hinder the departure of corrosion products from the of Al surface[4,6,20–22], high concentration also increases the cost against the original intention of employing low-cost Al current collectors.

From the anode side, another failure mechanism of cycling life results from lithium loss caused by the formation of a SEI and

[1]Beijing National Laboratory for Condensed Matter Physics, Institute of Physics, Chinese Academy of Science, 100190 Beijing, China. [2]Center of Materials Science and Optoelectronics Engineering, University of Chinese Academy of Sciences, 100049 Beijing, China. [3]Yangtze River Delta Physics Research Center Co. Ltd, 213300 Liyang, China. ✉e-mail: suoliumin@iphy.ac.cn

hydrogen evolution reaction (HER), which lead to a rapid capacity degradation. Typically, SEI formation in traditional nonaqueous LIBs irreversibly consumes 6%–15% Li ions, which are extracted from the cathode material, and reduces the specific energy by 5%–20% during the initial cycle[30,31]. However, as the extra HER that consumes the Li resource does not contribute to SEI formation, irreversible Li consumption is expectedly more severe in ALIBs[5,32]; this condition results in a high P/N (cathode-to-anode mass ratio)>1.5, which offsets the lithium compensation for the capacity loss from HER and SEI.

Based on the above information, the critical challenge in ALIB application is the stabilization of the Al current collector in low-concentration aqueous electrolytes. Here, we proposed a prolonged life strategy for aqueous batteries that involves the electrochemical regulation of the corrosion and passivation of Al to make it an electron supplement and a stable current collector. We used the hydrolyzation type-anodic additive (HTA) to achieve electrochemical–chemical passivation of the Al current collector and successfully precipitated $Al(OH)_3$ as a passivating layer on the Al surface. In addition, the passivation process, which involves Al oxidation at the cathode, can supplement electrons to anode, which promotes the compensation of the capacity loss resulting from hydrogen evolution at the anode side. The oxidation of Al is an electron supplement process, with an additional lithium source originating from the electrolyte as supplement. Therefore, in this context, an "electron supplement" was used instead of "lithium supplement." Furthermore, considering practical industrial applications, we designed a prototype of self-prolonging ALIBs (SP-ALIBs) $(Al‖LiMn_2O_4‖TiO_2)$ with an Al-sacrificing electrode, which enabled self-lithium supplementation via periodic switching between either full $LiMn_2O_4‖TiO_2$ or $Al‖TiO_2$ in combination with electrochemical regulation.

## Results

### Design of Al-passivating additives in an aqueous electrolyte

Figure 1 illustrates the schematic of our proposed HTA-passivating mechanism for Al current collectors in aqueous LiTFSI electrolytes. As reported previously[18,26], TFSI⁻ aggressively destroys the original $Al_2O_3$ layer to form dissolved $[Al(TFSI)_x^{3-x}]$ at the initial active spots of the Al surface film $(TFSI^-(aq) + Al_2O_3 \rightarrow Al(TFSI)_x^{3-x}(aq) + O_2 + e^-)$. Then, more active spots emerged, which caused pitting corrosion $(Al(s) - e^- + TFSI^-(aq) \rightarrow Al(TFSI)_x^{3-x}(aq))$. Therefore, given the inevitability of attacks from TFSI⁻ to the original passivation $Al_2O_3$ layer, our design principle was aimed at the chemical conversion of the initial electrochemical corrosion product to insoluble solid-phase aluminum compounds and coverage of the Al surface as a passivation layer. Previous

literature proposed the protection of Al from chloride corrosion through the construction of inorganic passivation layers[33,34]. Given this information, some insoluble aluminum compounds $(Al_aM_b)$, such as $Al(OH)_3$, can serve as target passivation products due to their relatively low solubility product constant; studies have reported the corrosion resistance of these compounds against chloride ions[34,35]. We applied the universal corrosion principles in aqueous batteries to prevent the Al current collector from being corroded by TFSI⁻. To form a robust and dense $Al_aM_b$ layer and restore the passivation layer on the Al surface, we contemplated on the use of Al ions generated by electrochemical corrosion during battery charging $(Al(s) - e^- \rightarrow Al^{3+})$ to chemically transform Al into solid-phase depositions and form a passivation layer (reaction path: $Al(s) - e^- + H_xM^{n-}(aq) + H_2O \rightarrow Al(OH)_3(s) + H_{x+1}M^{1-n}(aq))$ instead of dissolution in electrolytes as $[Al(TFSI)_x^{3-x}]$. First, electrochemical corrosion occurs, followed by chemical passivation that is, electrochemical–chemical passivation. Based on preceding findings, slightly soluble lithium weak acid salts, such as $Li_3PO_4$, $Li_2CO_3$, $Li_2SiO_3$, and $LiAlO_2$, were proposed as passivation additives for Al in aqueous electrolytes, whose anions after hydrolysis can combine with proton. These salts also perform a function similar to pH buffers in proximity to the Al foil and facilitate the formation of $Al(OH)_3$. Supplementary Information Note 2 provides a detailed discussion on the role of HTA.

To verify the anticorrosion effectiveness of HTA, we performed CA experiments using LiTFSI solution on three-electrode devices (Al-WE‖Ag/AgCl-RE‖Al-CE), where the potential of Al-WE was set at 4.5 V vs Li/Li⁺ for 10 minutes and then relaxed to the open-circuit potential (OCV) for 1 minute, for a total of 12 cycles (Fig. 2A). This periodic switching between 4.5 V and OCV eliminated the effect on transport limitations at the interface, which ensured that the experiment proceeded under harsh conditions. HTAs $(Li_3PO_4, Li_2CO_3, LiAlO_2,$ and $Li_2SiO_3)$ were added to 1 and 10 m LiTFSI solutions, whose content was saturated (<0.05 m). Al in the blank electrolyte (1 m LiTFSI) had a corrosion current density extremely higher than $10^{-3}$ A cm⁻², which indicates its severe corrosion in the diluted solution (Fig. 2A). Notably, after the introduction of saturated $LiAlO_2$, $Li_2CO_3$, and $Li_3PO_4$, the corrosion current density dropped dramatically by nearly three orders of magnitude $(10^{-6}$ A cm⁻²), which indicates a satisfying anticorrosion effectiveness. The CA experiments (Fig. 2A, B) were repeated twice to verify our claims (Figs. S7 and S8, respectively). Moreover, Supplementary Information Note 2 discusses the influence of the HTA amount on corrosion resistance.

To further support our claim, we used inductive coupled plasma (ICP) to detect the dissolved Al ion content in the electrolyte and

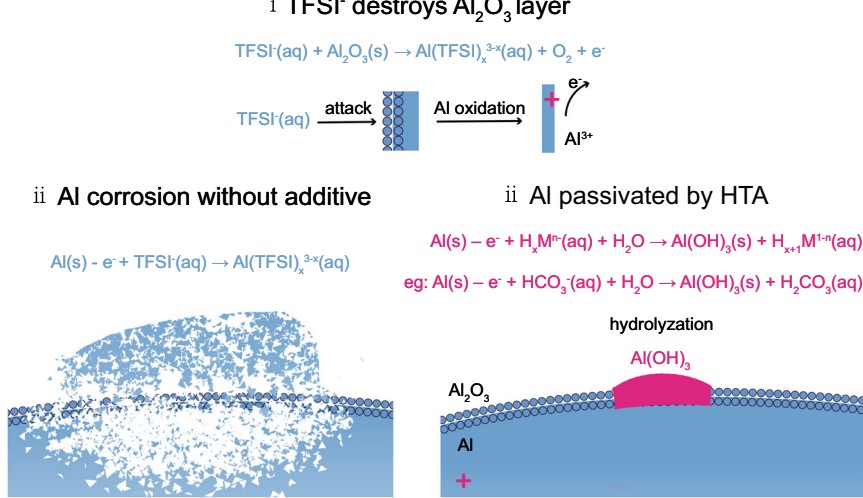

**Fig. 1 | How TFSI⁻ corrodes Al and HTA passivates Al.** Schematic of HTA-passivating Al mechanism in an aqueous LiTFSI electrolyte.

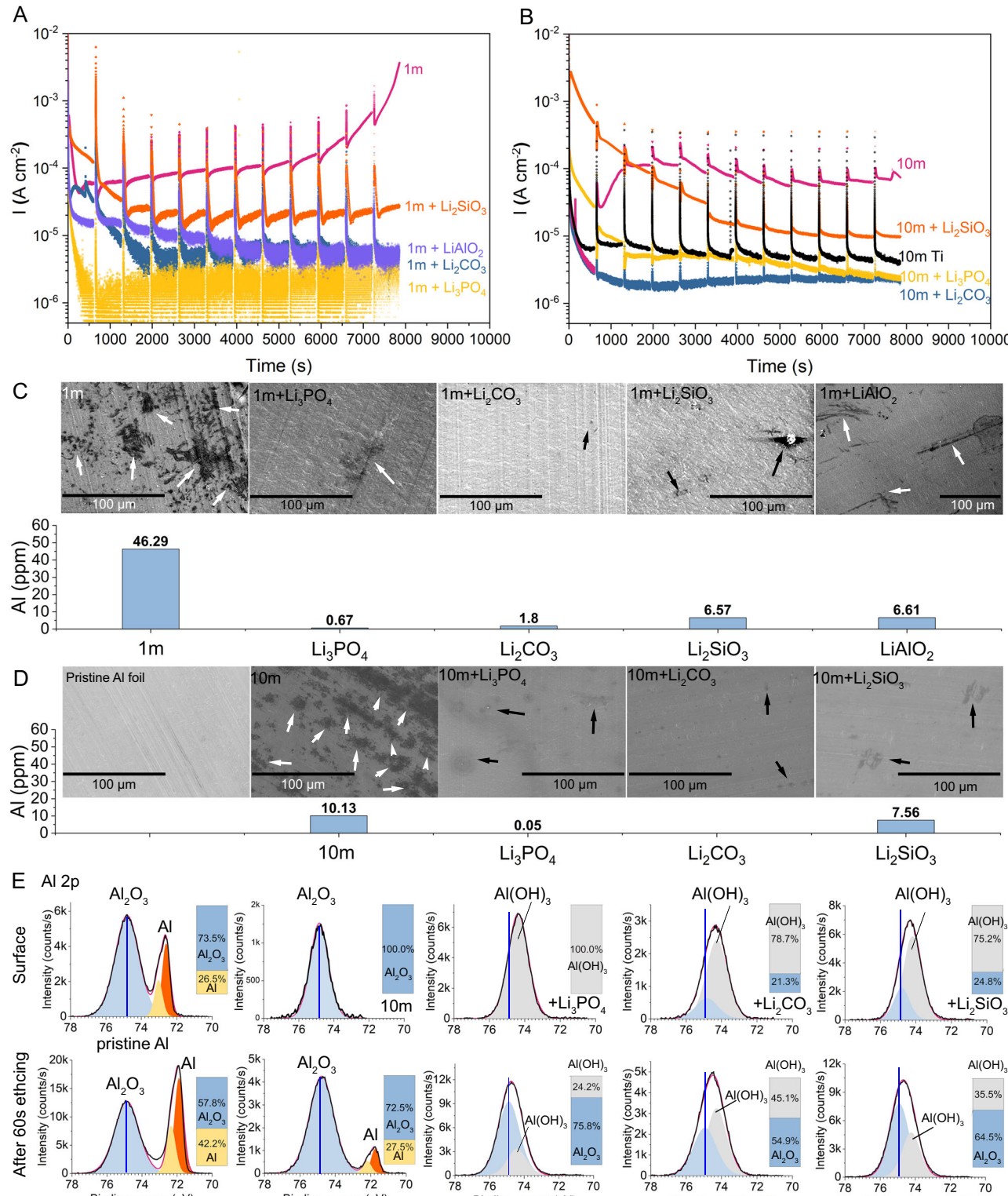

**Fig. 2 | Anticorrosion effect of HTA on the Al current collector.**
**A**, **B** Chronoamperometry (CA) experiments in 1 and 10 m LiTFSI solutions with saturated HTA (Li$_3$PO$_4$, Li$_2$CO$_3$, LiAlO$_2$, and Li$_2$SiO$_3$). **C**, **D** Al contents in electrolytes obtained via inductive coupled plasma (ICP) measurement and SEM images of Al foils after CA experiments using 1 and 10 m LiTFSI solutions with HTA. **E** Al 2$p$ X-ray photoelectron spectroscopy (XPS) spectra of Al foil after CA experiments using 10 m LiTFSI with HTA before and after 60 s Ar$^+$ sputtering. The blue vertical line is based on the Al$_2$O$_3$ peak of pristine Al.

scanning electron microscopy (SEM) to detect corrosion damage on the morphologies of Al foil after CA experiments (Fig. 2C, D). The determined Al contents of the electrolyte with Li$_3$PO$_4$ or Li$_2$CO$_3$ reached 0.67 and 1.8 ppm, respectively, which are considerably lower than that of in blank 1 m LiTFSI (46.29 ppm). In addition, the corrosive

pits were scarce and tiny compared with those of Al foil in 1 m LiTFSI. Furthermore, the cycled Al foils in 1 m LiTFSI with additives were washed in ethanol to prevent residual electrolyte from occupying their surface and immersed in 1 m LiTFSI to repeat the CA experiment under the same condition. The foils still exhibited an excellent anticorrosion

effect with a low corrosive current density (Figure S3) and low content of dissolved Al ($Li_3PO_4$: 0.46 ppm, $LiAlO_2$: 0.73 ppm, $Li_2CO_3$: 1.19 ppm, and $Li_2SiO_3$: 2.26 ppm; Fig. S4). Thus, a passivation layer formed in the first round of the CA experiment by HTA. Moreover, the passivation layer provided a robust and continuous protection to the Al foil in pure 1 m LiTFSI.

Additional corrosion experiments were conducted in 10 m LiTFSI to align with the concentrations used in actual battery systems. Ti serves as a highly corrosion-resistant current collector in aqueous batteries[36]. Compared with Ti, Al foils with $Li_3PO_4$ ($2.35 \times 10^{-6}$ A $cm^{-2}$) and $Li_2CO_3$ ($2.71 \times 10^{-6}$ A $cm^{-2}$) exhibit the same order of magnitude of corrosion current density as Ti ($5.34 \times 10^{-6}$ A $cm^{-2}$; Figs. 2B and S8). $Li_2CO_3$ and $Li_3PO_4$ are considered much better anti-corrosion additives than $Li_2SiO_3$ ($3.56 \times 10^{-5}$ A $cm^{-2}$), whose corrosion current density is lower by one order of magnitude than $Li_2SiO_3$. Evidently, the ICP revealed that the 10 m LiTFSI with $Li_2CO_3$ failed to detect the Al signal due to lower value than the detection minimum. The value for $Li_3PO_4$ was 0.05 ppm, which is considerably lower than that of $Li_2SiO_3$ (7.56 ppm). Such result was probably due to the substantial change in pH from 6.82 (10 m LiTFSI), 6.87 (10 m LiTFSI + $Li_2CO_3$), 6.93 (10 m LiTFSI + $Li_3PO_4$) to 9.20 (10 m LiTFSI + $Li_2SiO_3$; Fig. S2). Thus, in the following section, we focused on the HTAs ($Li_2CO_3$ and $Li_3PO_4$) in 10 m LiTFSI and evaluated their electrochemical performance.

XPS was conducted to detect the passivation layer that formed on the Al foils after the introduction of HTA (Fig. 2E). Considering the insulating $Al_2O_3$ on pristine Al foils, we used its XPS spectrum as a reference ($Al_2O_3$, Al 2$p$: 74.81 eV). $Al(OH)_3$ was detected at 74.37 eV for $Li_3PO_4$, 74.27 eV for $Li_2CO_3$, and 74.19 eV for $Li_2SiO_3$ (Fig. 2E), consistent with the reported value in literature (74.4 eV)[37]. This finding indicates that the $Al^{3+}$ originating from the oxidization of Al interacted with the anions of HTAs ($HPO_4^{2-}$, $H_2PO_4^-$, $HCO_3^-$, and $HSiO_3^-$) and was hydrolyzed to insoluble solid $Al(OH)_3$ to form the passivation film that covered the active corrosion spots on the Al surface film and blocked further corrosive reaction. To further clarify our claims, we compared the pristine samples with their corresponding $Ar^+$ sputtering samples. The blank 10 m LiTFSI showed asymmetric double Al 2$p$ peaks belonging to the metallic Al foil, which indicates the destruction of pristine $Al_2O_3$ film and its failure to exert anticorrosion action against the aqueous LiTFSI electrolyte. By contrast, for the HTAs observed after 60 s Ar ion sputtering, signals from the passivation layers $Al(OH)_3$ weakened, but remained with a relatively high intensity. Specifically, during etching, the signals of pristine $Al_2O_3$ intensified, which signifies the presence of double passivation layers consisting of an $Al(OH)_3$ top layer and a pristine $Al_2O_3$ bottom layer. Thus, ceaseless destruction of the $Al_2O_3$ layer by $TFSI^-$ was impossible due to the passivation of Al foil because once the corrosion pitting sites appeared, the dissolved $Al^{3+}$ rapidly reacted with the HTA, which resulted in $Al(OH)_3$ layer covering the undamaged $Al_2O_3$ layer around the pitting site. Based on the information mentioned above, a compact and stable passivation layer $Al(OH)_3$ formed on the surface of Al foil. Moreover, we observed our proposed mechanism in aqueous sodium-ion electrolytes, in which $Na_2CO_3$ exhibited an effective anticorrosion effect on the Al foil (Figs. S8−S10). Thus, our concept is universal and applies to the stabilization of Al foils in aqueous batteries.

### Identification and quantification of Al oxidation corrosion for electron supplement

During corrosion, the current collector provided additional irreversible overcharging capacity as a result of Al oxidation. In consideration of the charge balance of batteries, Al oxidation at the cathode side corresponded to the reduction on the anode side, which balanced the redox reaction. Thereby, in such a redox reaction, the prelithiation of the anode compensates for the extra electron loss on the cathode side. To verify the life extension mechanism via Al oxidation, we evaluated individual Al and Ti current collectors in a full cell (LMO || 10 m LiTFSI ||

TiO, P/N = 2) (Fig. 3). An extra 2 $cm^2$ bare Al foil was attached behind the 0.79 $cm^2$ $LiMn_2O_4$ electrode to amplify electron supplementation via the oxidation of Al as a prelithiation additive. Figure 3 reveals that compared with the Ti current collector, the battery with an Al current collector exhibited an extra plateau of around 1.2−1.3 V at the first charge process, and such finding was attributed to the corrosion of the Al current collector. Moreover, the extra plateau of approximately 1 V at the following discharge process belongs to the overlithiation of $LiMn_2O_4$. This result indicates the corrosive oxidation of the Al current collector at the earlier stage of the first charge, accompanied with the intercalation of Li ions in $TiO_2$. This condition resulted in the partial prelithiation of the anode ($Li_xTiO_2$) and overlithiation of $Li_{1+x}Mn_2O_4$ in the following discharge (Fig. 3B). Overall, Al was oxidized into Al ions that were dissolved in the electrolyte. In addition, Li ions from the electrolyte were intercalated in the active material. The electrolyte served as an additional source of lithium, and oxidation of the Al foil was the extra electron supplement. Thus, oxidation of Al current collectors at the cathode side can function as a prelithiation additive to prolong the life of batteries.

Despite the prelithiation via the oxidation of the current collector, the feasibility of electrochemical regulation of corrosion and passivation requires further verification before a proof of concept can be proposed. Suppose the corrosion process cannot be controlled through electrochemical regulation. In such a case, the electrode will be damaged, and continuous irreversible overcharging will overcompensate for the fully lithiated $Li_{0.5}TiO_2$, which will result in the low utilization of capacity and further acceleration of other parasitic reactions, such as HER. Such condition will lead to the degradation of the battery's lifespan. Fortunately, this problem can be solved with the use of HTA. A small amount of Al has to be corroded first to offer some dissolving Al ion as the reaction source for the following passivation by HTA. Such a mechanism ensures the electrochemical regulation of the corrosion−passivation of Al for the long-term cycle life of battery via the stabilization of the Al current collector and appropriate supplementation of lithium loss.

To further confirm that the oxidation of Al only prelithiates the anode in the initial charge process and the passivated Al by HTA serves as a robust current collector in the following cycle, we designed three-electrode cells with two fresh $TiO_2$ anodes to distinguish Al oxidation in the first and second cycles and an excess $LiMn_2O_4$ cathode with a high P/N ≈ 3 to provide sufficient lithium resource to sustain the

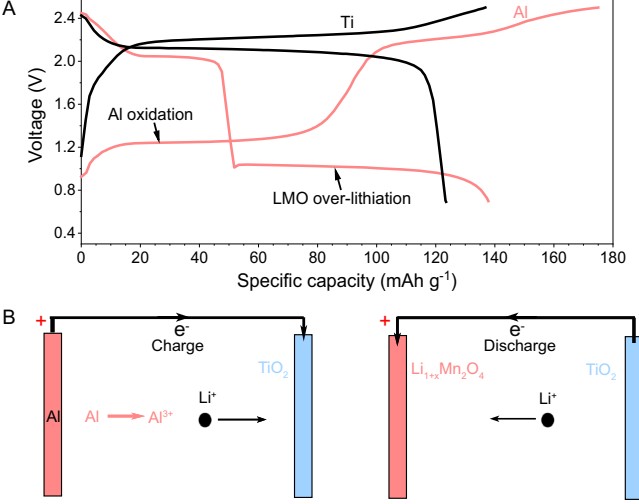

**Fig. 3 | Identification of Al oxidation corrosion for electron supplement. A** First charge−discharge profiles of $LiMn_2O_4$−$TiO_2$ cells with Al or Ti current collector in 10 m LiTFSI. **B** Schematic of the prelithiation of $TiO_2$ via Al current collector oxidation in the charge process and overlithiation of $LiMn_2O_4$ in the discharge process.

double $TiO_2$ anodes (Fig. 4A). Accordingly, the charging program of batteries was set up in the following order: (1) cell configuration: ①LMO electrode /②$TiO_2$ electrode, first full charging at 1 C; (2) cell configuration: ①LMO electrode /③$TiO_2$ electrode, second full charging at 1 C rate. As the entire process occurred without discharging, interference was eliminated from overlithiation of $LiMn_2O_4$. As shown in Fig. 4B, regardless of whether $Li_3PO_4$ and $Li_2CO_3$ was used, the Al oxidation occurred at approximately 1.2 V on the first charge and almost disappeared on the second charge ($Li_2CO_3$: 1st 59.34 mAh g$^{-1}$ and 2nd 2.73 mAh g$^{-1}$; $Li_3PO_4$: 1st 77.22 mAh g$^{-1}$ and 2nd 9.68 mAh g$^{-1}$). Thus, the corrosion and passivation with the aid of additives mainly transpired during the first charge, accompanied by the electron supplement from the Al oxidation. In the following cycles, the Al current collector was stabilized via passivation to support the long-term calendar life of aqueous batteries.

## Effectiveness of HTA in prolonging the cycle life of batteries

We assembled $LiMn_2O_4$||$TiO_2$ full cells (P/N ≈ 2) to evaluate the effectiveness of $Li_3PO_4$ and $Li_2CO_3$ additives (Fig. 5). The Al and Ti current collectors with a blank electrolyte consisting of 10 m LiTFSI served as control samples. To magnify the passivation process and the effects of electron supplement from Al oxidation, we attached an extra Al foil (2 cm$^2$) behind the $LiMn_2O_4$ electrode (0.79 cm$^2$), excluding the Ti current collector. As shown in Fig. 5A, Al corrosion commenced at approximately 1.2 V, along with Al passivation by HTA, which reflects an increase in the plateau from 1.2 V to 1.4 V, consistent with the peak of ~1.3 V in the dQ/dV plot (Fig. 5B). Furthermore, full cells can survive in the relatively low-concentration electrolyte (10 m LiTFSI) with HTAs, whose cycle life improved substantially from 12.6% to above 84.7% ($Li_2CO_3$) and 89.1% ($Li_3PO_4$) after 1500 times. Such result was due to the

suppressed continuously irreversible Al corrosion via passivation, which maintained the good electronic conducting contact between the electrode and current collector. In addition, the cycling stability of anticorrosion Ti (65.2%, 1500 cycles) cannot surpass that of Al with HTA, which confirms the effect of the prelithiation of $TiO_2$ during electrochemical–chemical Al passivation. Figure 5D indicates a clear distinction between the efficiency of cells with and without additives. Batteries with HTA showed a rapid increase in efficiency at the initial cycles, with 1500 cycles average values of 99.36% ($Li_2CO_3$) and 99.37% ($Li_3PO_4$), which are considerably higher than that of 10 m LiTFSI (98.03%). Thus, the electrochemical–chemical Al passivation derived by HTA can stabilize the Al current collector in relatively low-concentration electrolytes and offer an electron supplement for the prolonged cycle life of ALIBs.

## Prototype of SP-ALIBs

From a practical point of view, as the mass loading of commercial Li-ion batteries reaches above 20 mg cm$^{-2}$. Given the limited reactive mass of the Al current collector, the effectiveness of electron supplement from the Al current collector alone may be extremely low in actual conditions. To solve this problem, we designed a new battery type through introduction of an Al sacrificial prelithiation electrode dedicated to being oxidized as the electron supplement (Fig. 6B). The detailed program is as follows. Prior to cycling, the cell was precharged with an Al sacrificial prelithiation electrode to obtain $TiO_2$ anode partially prelithiated (Al||$TiO_2$). The prelithiation capacity of $TiO_2$ is depicted with the red short bar in Fig. 6A. Subsequently, the $LiMn_2O_4$ electrode, instead of the sacrificial Al electrode, initiated the cycling process ($LiMn_2O_4$||$TiO_2$). As shown in Fig. 6A, when $TiO_2$ was fully lithiated by $LiMn_2O_4$, the quantity of Li extracted from $LiMn_2O_4$

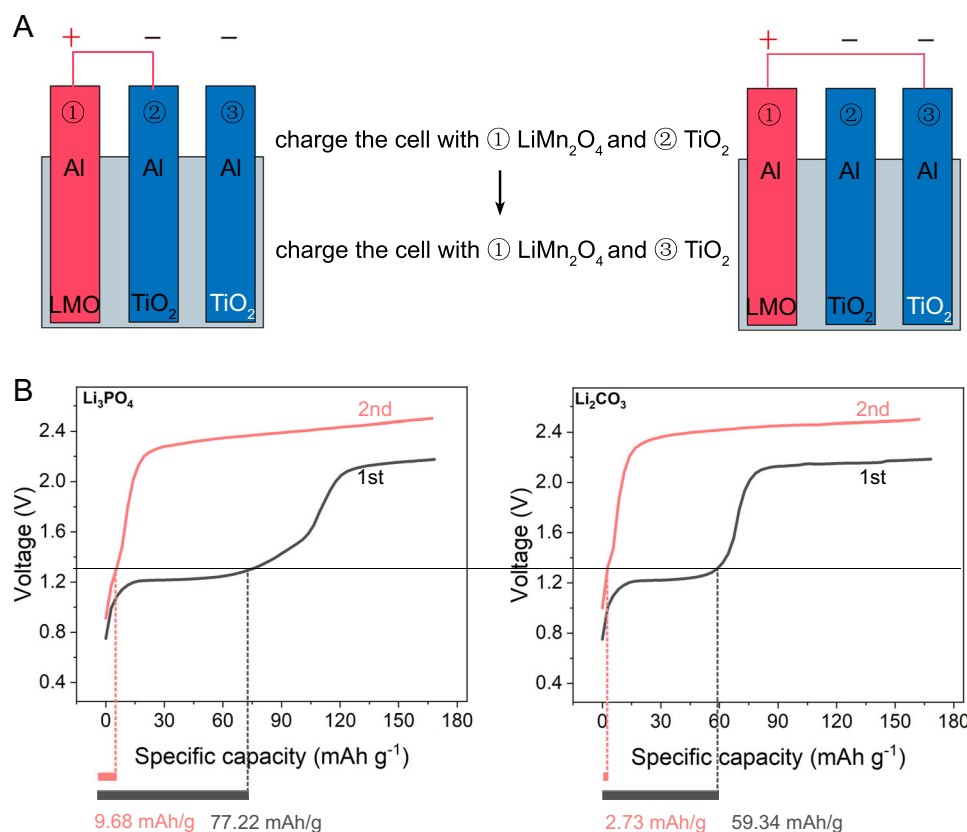

**Fig. 4 | Duration of Al oxidation corrosion for electron supplement. A** Process diagram for continuous charging of dual-anode battery. Charge the battery with ① $LiMn_2O_4$ cathode and ② $TiO_2$ anode, then charge it with ① $LiMn_2O_4$ cathode and ③ $TiO_2$ anode. **B** Consecutive charging profiles of batteries with one $LiMn_2O_4$ cathode and two $TiO_2$ anodes in 10 m LiTFSI + $Li_2CO_3$ or $Li_3PO_4$.

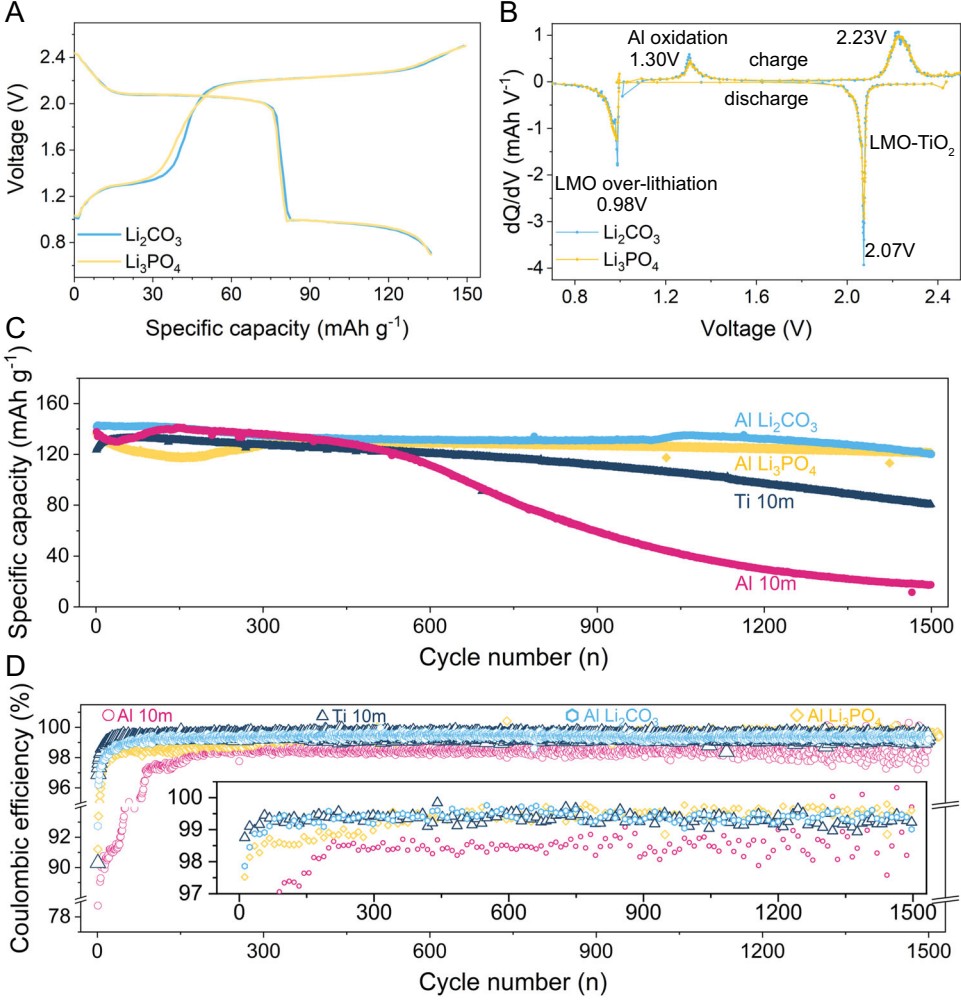

**Fig. 5 | Cycling performance of ALIBs with Li₂CO₃ and Li₃PO₄ additives. A**, **B** First charge–discharge profiles and dQ/dV curves of LiMn₂O₄ ||TiO₂ cells with an Al current collector in 10 m LiTFSI with additives. **C** Cycling stability of LiMn₂O₄ ||TiO₂ cells with Al and Ti current collectors in 10 m LiTFSI with/without additives. **D** Comparison of coulombic efficiency with a magnified view inside.

during the charging process was less than the Li extracted from TiO₂ during the discharging process, which caused the overlithiation of LiMn₂O₄. Overall, Al is a controllable electron supplement. In addition to the initial stage, when capacity faded after long cycles, the cathode can be switched to an Al electrode for another electron supplementation. Preliminarily, we constructed a pouch cell (Al|| LiMn₂O₄-Ti||TiO₂) employing a Ti current collector to clarify the extra electron supplement only from the sacrificial prelithiation Al electrode (Figure S15). The anode was initially prelithiated using a sacrificial prelithiation Al electrode (2.4 V constant potential between Al and TiO₂ for 20 minutes at 25 °C), which corresponded to the new discharge plateau of 0.98 V with an overlithiation capacity of 26.7 mAh g⁻¹) and subsequently resulted in notably better cycling performance. Finally, we constructed a 0.5 Ah multilayer stacked SP-ALIBs with Al foil as the current collector (Al ||LiMn₂O₄-Al ||TiO₂, 0.5 Ah, and 55.2 Wh kg⁻¹), whose loading mass was 20 mg cm⁻² for LiMn₂O₄ and 12 mg cm⁻² for TiO₂, which are close to the level of commercial Li-ion batteries. The introduction of the sacrificial Al electrode lowered the energy density. However, as the sacrificial Al electrode had a high theoretical capacity of 2980 mAh g⁻¹, the overall estimated energy density decreased by less than 5%. After the prelithiation of TiO₂ anode by the Al electrode (2.8 V constant potential between Al and TiO₂ for 22 h at 35 °C), the discharge profile of the initial cycle showed the overlithiation plateau of LiMn₂O₄ (Fig. 6C). This finding reveals that high voltage and temperature cause the

partial destruction of the passivation layer on sacrificial prelithiation Al electrodes generated by HTA and the feasibility of using applied potential and temperature to regulate Al corrosion with time. For the simultaneous protection of the Al current collector and utilization of Al corrosion for electron supplement, an additional sacrificial Al electrode was used, and the operating conditions of the Al current collector and sacrificial electrode were differentiated. In this case, the Al current collector was protected from HTA and functioned well throughout the battery lifespan. Meanwhile, the sacrificial Al electrode was oxidized in harsh conditions to prelithiate TiO₂. The capacity retention of SP-ALIBs with HTA (Li₂CO₃) rose to 70.1% after 200 cycles, which is considerably higher than that of the traditional LiMn₂O₄ ||TiO₂ cell without HTA (49.5%) (Fig. 6D).

Furthermore, a distinct coulombic efficiency difference was observed between SP-ALIBs (average 99.32% for 200 cycles) and normal cells (93.14%). The low efficiency was attributed to the failure of the Al current collector at the cathode and electrolyte pollution by Al³⁺. Thus, the HTA must convert the corrosion product Al³⁺ to precipitate it as a passivation layer. We validated the reliability of our design through the assembly and evaluation of three SP-ALIBs (0.5 Ah) to (Fig. S17), which showed an excellent consistency. The prototype of SP-ALIBs achieved the stability of Al current collectors in harsh aqueous environments and supplemented extra electrons via the sacrificial prelithiation Al electrode, which has great potential for industrial application. An open design for electrolyte replenishment should be

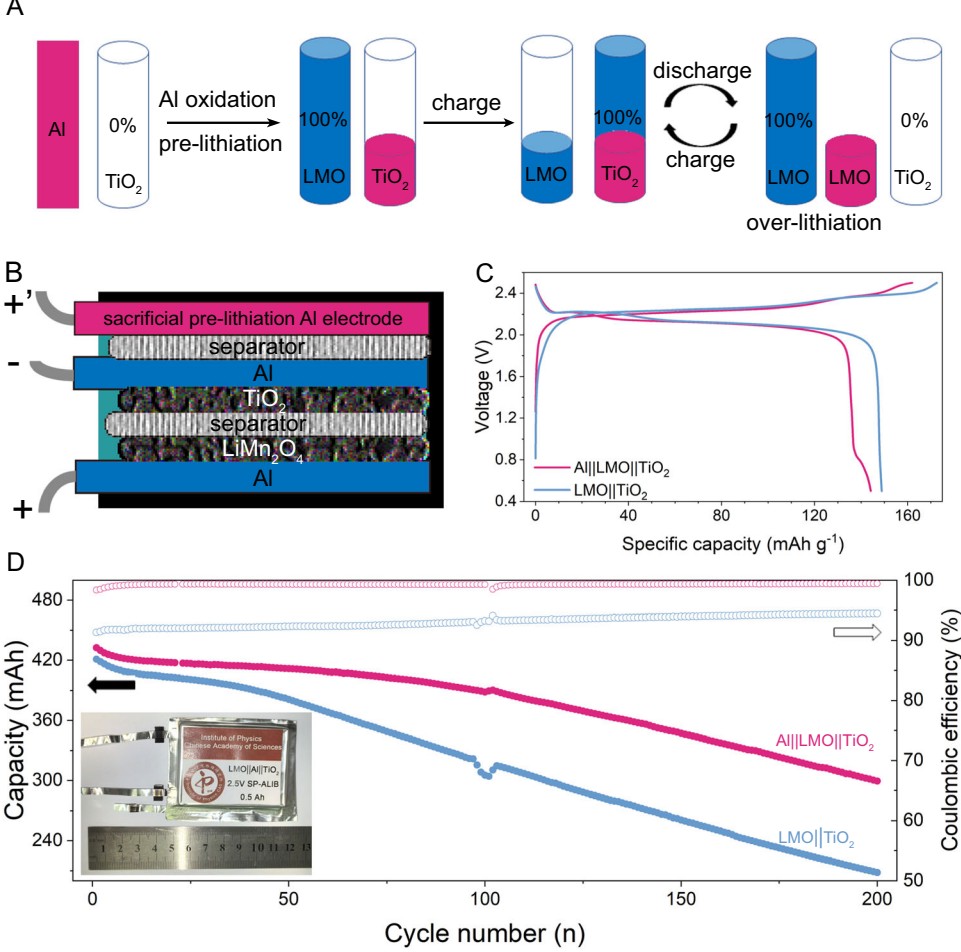

**Fig. 6 | Prototype of SP-ALIBs (Al||LiMn$_2$O$_4$||TiO$_2$). A** Program for SP-ALIBs. **B** Schematic of SP-ALIBs. **C** First charge–discharge profiles of the formation process of the 0.5 Ah SP-ALIB at 0.8 C after the prelithiation of TiO$_2$ anode at 2.8 V for 22 h via the sacrificial prelithiation Al electrode and traditional pouch cell. **D** Comparison of cycle stabilities of SP-ALIBs and traditional pouch cell at 1 C rate. Inset: photo of a practical SP-ALIB.

considered for practical purposes due to electrolyte loss during long-term cycling.

## Discussion

In this work, the anticorrosion effectiveness of HTAs was discovered, and their mechanism was investigated. With the assistance of HTA, a robust Al(OH)$_3$ passivation layer formed on the Al surface, which dramatically lowered the corrosion current density by nearly three orders of magnitude (10$^{-6}$ A cm$^{-2}$) in 1 m LiTFSI solution. Such a condition made the Al foil corrosion-resistant and comparable to Ti. Moreover, through regulation by HTA, the controllable electrochemical corrosion process on the cathode side can be used to pre-lithiate the anode. With the robust Al current collector and extra electron supplemented by Al passivation, full cells can survive in relatively low-concentration electrolytes (10 m LiTFSI) with the employment of HTA, whose cycle life improved substantially from 12.6% to above 84.7% (Li$_2$CO$_3$) and 89.1% (Li$_3$PO$_4$) after 1500 times. Such finding implies the feasibility of using Al corrosion–passivation regulation to prolong the life of aqueous batteries. Al passivation ensured that the current collector's long-term chemical/electrochemical stability and Al oxidation corrosion compensated for the irreversible capacity loss caused by HER and SEI formation. Based on these results, we designed a prototype of SP-ALIBs with an extra Al sacrificial prelithiation electrode as an electron supplement. The capacity retention of the 0.5 Ah prelithiated SP-ALIBs with HTA (Li$_2$CO$_3$) rose to 70.1% at 1C after 200 cycles, and such value is higher than that of the traditional LiMn$_2$O$_4$ || TiO$_2$ cell (49.5%). The average efficiency increased from 93.14% to 99.32%. The discovery of HTA in a broader context revealed the once-overlooked complicated corrosion and passivation processes that occur in batteries in general. Through exploration of anticorrosion additives, the SP-ALIBs will gain more potential to achieve long-life ALIBs, which benefits the sustainable and lifelong industrial applications of ALIB.

## Methods

### Preparation of LiMn$_2$O$_4$-TiO$_2$, LiMn$_2$O$_4$-TiO$_2$-TiO$_2$, Al-TiO$_2$, and LiMn$_2$O$_4$-Al-TiO$_2$ pouch cells

All pouch cells other than 0.5 Ah were assembled using a glass fiber as a separator, 10 m LiTFSI as the electrolyte (some with HTA), and Al plastic film as packaging. We fabricated the LiMn$_2$O$_4$ cathode by mixing LiMn$_2$O$_4$, carbon black (CB), and polyvinylidene fluoride (PVDF) at a weight ratio of 8:1:1 in N-methyl pyrrolidinone (NMP) in an SK-300SII CE mixing machine (SHASHIN KAGAKU Co., Ltd.) for 20 minutes at a speed of 2000 rsm to produce a black slurry. The slurry was spread uniformly on a clean Al or Ti foil and then dried. TiO$_2$ anode was prepared using the same process and spread on the Al foil using the same CB, PVDF, and NMP at a weight ratio of 8:1:1. An extra 2 cm$^2$ bare Al foil was attached behind the Al current collectors of LiMn$_2$O$_4$ in Figure 345 aside from LiMn$_2$O$_4$ with Ti current collector. The small pouch cells included unilaterally coated LiMn$_2$O$_4$ cathode (diameter: 10 mm, thickness: ~41.2 μm, and mass loading: ~3.2 mg cm$^{-2}$), unilaterally coated TiO$_2$ anode (diameter: 10 mm, thickness ~29 μm, and

mass loading ~1.6 mg cm$^{-2}$), and separator (area 2.5 cm × 1.3 cm; thickness: ~55 μm). The 10 m LiTFSI was used as an electrolyte. HTA was added to the electrolyte as an additive until saturation and the supernatant was used in the pouch cells. $Li_2CO_3$, $Li_3PO_4$, $Li_3SiO_3$, or $LiAlO_2$ was hard to dissolve in 10 m LiTFSI, and saturation was reached at a concentration below 0.05 m.

The 0.5 Ah SP-ALIBs were assembled with eight layers of $LiMn_2O_4$ cathode, seven layers of $TiO_2$ anode, and a sacrificial Al electrode. $LiMn_2O_4$ and $TiO_2$ had mass loadings of ~20 (90 wt%) and ~12 mg cm$^{-2}$ (80 wt%), respectively. The cathode area was 4.5 cm × 5.8 cm, that of the anode was 4.3 cm × 5.6 cm, and the width of the PP separator was 6 cm. The sacrificial Al electrode (4.5 cm × 5.8 cm) was sanded using a sandpaper to facilitate corrosion and placed on one side of the cell. Prior to standard formation, the sacrificial Al electrode prelithiated the cell at a constant voltage of 2.8 V at 35 °C for 22 h.

The electrolyte in this study was consistently expressed in terms of molality (m).

### Electrochemical measurements

CA experiments were conducted in a 25 °C constant-temperature chamber using CHI 660E electrochemical workstation in three-electrode cells with 10 mL electrolyte, and the area of the Al working electrode was 1 cm². The CA experiments on LiTFSI solution were performed in three-electrode devices (Al-WE ∥Ag/AgCl-RE ∥Al-CE), where the potential of the Al-WE was set at 4.5 V vs Li/Li$^+$ for 10 minutes, relaxed to the OCV for 1 minutes, and cycled for 12 times as above. The batteries were tested using a LAND battery test system (Wuhan, China). The pouch cells in Figs. 3 and 5 were tested at the 3 C rate, and those in Figs. 4 and 6 were tested at a 1 C rate. The small pouch cells in Figs. 3, 4, and 5 had a cutoff voltage from 0.7 V to 2.5 V. The 0.5 Ah pouch cells in Fig. 6 experienced a cutoff voltage of 0.5 V to 2.5 V. All batteries were tested in a 25 °C constant-temperature chamber. Calculations of the specific energy of 0.5 Ah pouch cells was based on the masses of the cathode, anode, separator, and electrolyte.

### Characterizations

SEM images of the morphologies of the Al foil after CA experiments were obtained using a Hitachi S-4800 field-emission scanning electron microscope operated at 10 kV. XPS analysis was performed using an ESCALAB 250 Xi, ThermoFisher with Mg/Al Kα radiation. All the binding energies were referenced to the C 1$s$ line at 284.8 eV. All electrolyte contents were measured on an ICP-OES Agilent 5800.

### Reporting summary

Further information on research design is available in the Nature Portfolio Reporting Summary linked to this article.

## Data availability

Source data are provided in this paper. Extra data are available from the corresponding author upon reasonable request. Source data are provided with this paper.

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

## Acknowledgements

This work was supported by the National Key Research and Development Program of China (2022YFB2404500), the National Natural Science Foundation of China (U22B20124), CAS Youth Interdisciplinary Team, and the Center for Clean Energy.

## Author contributions

L.S. conducted the project. B.L. and L.S. conceived the concept and designed this work. B.L. and L.S. wrote the paper. B.L. conducted the electrochemical experiments, SEM, and ICP measurements. T.L. (Tianshi Lv) conducted the XPS measurements. T.L. (Tianshi Lv), A.Z. and X.Z. participated in constructing the 0.5Ah batteries. Z.L. participated in drawing the schematic diagram. T.L. (Ting Lin) assisted in analyzing the composition of the passivation layer. All authors participated in the analysis of the results and reviewed the manuscript.

## Competing interests

The authors declare no competing interests.
