## [Peer Review File · Nature Communications]

Aluminum Corrosion-Passivation Regulation Prolongs Aqueous Batteries LifeREVIEWER COMMENTS

Reviewer #1 (Remarks to the Author):

In the present work, Suo and co-workers report the utilization of hydrolyzation-type anodic additives (HTA) and precipitation-type anodic additives (PTA) to tame the corrosion issue faced by aqueous batteries. The topic is of vital importance for the development of aqueous batteries and the approach proposed in this work is elegant and practically feasible. Overall, the manuscript is well-organized and the results are quite promising. In particular, the cyclability of the prototype cells could be substantially improved with the employment of HTA and PTA. Therefore, I would recommend the publication of this work in Nature Communications after addressing the following minor issues.

1. Line 59. "...highly dependent on the organic salts such as LiTFSI and LiOTF". The full expansions of LiTFSI and LiOTF need to be provided.
2. Lines 154-155. "Furthermore, the cycled Al foils mentioned above in 1m LiTFSI with additives were washed in ethanol and then put into 1m LiTFSI to repeat the CA experiment at the same condition." The authors are recommended to elaborate further why ethanol was selected for removing the surface products on cycled Al foils.
3. Lines 227-229. For the over-lithiation of LMO, the extra lithium source originates from the pre-lithiation of Li_xTiO_2 . However, it would be better if the author could explain further where the extra lithium comes from for the lithiation of TiO_2 during the early stage of the first charge.

Reviewer #2 (Remarks to the Author):

The manuscript is constructed based on three main logics; 1) an addition of Li_2CO_3 and Li_3PO_4 can inhibit Al corrosion, 2) a passivation process involving Al oxidation at the cathodes can serve as a lithium supplement to LiMn_2O_4 cathodes, thereby compensating for the capacity loss from HER at the anodes, 3) an introduction of a sacrificial electrode can retard Al corrosion. However, considering the significant scientific and performance concerns raised in this paper, this reviewer does not recommend its publication in Nature Communications.

General Concept Issues:

All three of the logics are neither new nor based on correct scientific consideration.

1. The concept introduced in this paper, aiming to suppress Al corrosion with carbonate and phosphate ions, has already been reported since the 1980s (Corrosion Science (1999), 41 (9) 1743-1767; [https://doi.org/10.1016/S0010-938X\(99\)00012-8](https://doi.org/10.1016/S0010-938X(99)00012-8), CORROSION (1994), 50(3), 205–214; <https://doi.org/10.5006/1.3293512>, Corrosion and Protection of Aluminium Alloys (1986), Moscow, Publishing House Metallurgy).

2. When bare Al is exposed to the electrolyte and oxidized to Al^{3+} ($\text{Al} \rightarrow \text{Al}^{3+} + 3\text{e}^-$), a simultaneous reduction of the electrolyte takes place on the Al surface. This corrosion reaction occurs at both the anodes and cathodes in aqueous batteries due to the low redox potential of Al/Al^{3+} . Importantly, this process does not induce any pre-lithiation reactions.

3. The introduction of a sacrificial electrode to retard Al corrosion was reported very early in the field of corrosion, and therefore, it has been applied commercially for a significant period of time.

<https://xapps.xyleminc.com/Crest.Grindex/help/grindex/contents/CathodicProtection.htm>

Mechanism and Performance Issues:

1. The Al₂O₃-destructive reaction mentioned in this manuscript (TFSI⁻ + Al₂O₃ → Al(TFSI)₃ + O₂ + e⁻) is considered to be highly unlikely. This reviewer believes that the Gibbs formation free energy of Al₂O₃ formation is actually larger than that of Al(TFSI)₃.

2. Insoluble compounds, such as Al₂O₃ and Al(OH)₃, invariably form in the aqueous electrolyte on the Al surface even without salts, and help to reduce the surface area of bare Al. This is a fact that is well-established and not novel. To suppress Al corrosion significantly, the rate of precipitation reactions should outpace the rate of Al oxidation reaction.

Furthermore, the precipitates should include minimal cracks and defects, and should firmly adhere to the Al surface. However, general Al compounds, including Al₂O₃, Al(OH)₃, and AlPO₄, are hard to meet these criteria, resulting in limited inhibition effectiveness (Please check the above literature in General Concept Issues 1). Consequently, the addition of Li₂CO₃ and Li₃PO₄ attempted in this study does not exhibit a marked suppression effect on Al corrosion (Figure 2a and 2b; a leakage current of several μA cm⁻² was observed at mild potentials of 4.5 V. The leakage current should remain below μA cm⁻² even at high potentials)

3. Figure 5: Cycling performance comparisons should be conducted at a slow C-rate (e.g. 0.1 C-rate).

4. Ti is widely employed as a cathode current collector due to its high stability in the aqueous electrolyte even at high potentials over 5 V (Journal of Solid State Electrochemistry, 26, 85-95, 2022). This reviewer could not understand why the leakage current from the Ti current collector is so high in this paper (maybe the purity issue of the Ti current collector used in this study).

Reviewer #3 (Remarks to the Author):

This is a very interesting work on the passivation of Al current collectors and their use in long-lifetime aqueous batteries. The additives are helpful in extending the cycle time and improve the coulombic efficiency. I would like to recommend its publication. The following comments should be addressed in the revision.

1. Figure 1 is a bit complicated to follow. The authors are suggested to make it clearer and more focused.

2. The passivation layer is the key. The main characterizations are by XPS. The references of the XPS peaks and the analysis should be provided in detail.

3. In Figure 5d, the coulombic efficiencies of the first few cycles are low. Why? I do not think the charge transfer to form the passivation layer is large enough to significantly lower the coulombic efficiency.

4. For Al||LiMn₂O₄||TiO₂ full cells in Figure 6, what are the mechanisms for capacity decay? The additives and passivation of Al current collectors improve the capacity retention by 17% over 200 cycles. A comparison of the cell impedance before and after the cycling should be

given.

Reviewer #4 (Remarks to the Author):

This paper describes a new method of passivating aluminium current collectors for use in aqueous Li-ion batteries, which has potentially significant implications for the commercial development of the technology.

However, I have a concern around the lack of information provided by the authors regarding the repeatability of their measurements. No mention is made of the number of repeat tests, there are no error bars in any of the plots and no statements of measurement uncertainty are provided. This must be addressed before publication can be considered.

Reply to the comments from Reviewer #1:

Comments: In the present work, Suo and coworkers report the utilization of hydrolyzation-type anodic additives (HTA) and precipitation-type anodic additives (PTA) to tame the corrosion issue faced by aqueous batteries. The topic is of vital importance for the development of aqueous batteries and the approach proposed in this work is elegant and practically feasible. Overall, the manuscript is well-organized and the results are quite promising. In particular, the cyclability of the prototype cells could be substantially improved with the employment of HTA and PTA. Therefore, I would recommend the publication of this work in Nature Communications after addressing the following minor issues.

Reply: We thank you for the reviewer's positive comments and constructive suggestions, which are very helpful in improving the quality of our manuscript.

Q1: Line 59. "...highly dependent on the organic salts such as LiTFSI and LiOTF". The full expansions of LiTFSI and LiOTF need to be provided.

R1: Thank you for your detailed comments. The full name and chemical formula of LiTFSI is Lithium bis(trifluoromethanesulfonyl)imide and $\text{LiC}_2\text{F}_6\text{NO}_4\text{S}_2$. The full name and chemical formula of LiOTF is lithium trifluoromethanesulfonate and $\text{CF}_3\text{LiO}_3\text{S}$. A corresponding correction has been made in the main text.

Main text "-----"

on the other hand, the formation of SEI in the neutral aqueous electrolyte is highly dependent on the organic salts such as bis(trifluoromethane sulfonyl)imide (LiTFSI) and lithium trifluoromethanesulfonate (LiOTF)^[3], whose anions are quite aggressive against Al no matter in non-aqueous and aqueous electrolyte^[19-21, 24-27],

-----"

Q2: Lines 154-155. "Furthermore, the cycled Al foils mentioned above in 1m LiTFSI with additives were washed in ethanol and then put into 1m LiTFSI to repeat the CA experiment at the same condition." The authors are recommended to elaborate further why ethanol was selected for removing the surface products on cycled Al foils.

R2: Thank you very much for your comment. Ethanol is chosen because it evaporates fast. Firstly, the Al foil must be washed after the first corrosion experiment in 1m LiTFSI with additives to avoid the residual electrolyte on the surface of the Al foil. After the electrolyte evaporates, LiTFSI and additives like Li_2CO_3 and Li_3PO_4 will precipitate and be introduced into the pure 1m LiTFSI electrolyte in the next corrosion experiment. Secondly, both water and ethanol can remove the electrolyte left on the surface of Al foil, and the difference is that ethanol evaporates much faster than water. After the immersion and flushing, Al foil must be dried first and used as a working electrode in the next corrosion experiment. A shorter drying time is better because the corroded Al foil gets passivated constantly in the air by generating an Al_2O_3 layer at the active spots.

Main text "-----"

Furthermore, the cycled Al foils mentioned above in 1m LiTFSI with additives were washed in ethanol to avoid the residual electrolyte on the surface of the Al foil and then put into 1m LiTFSI to repeat the CA experiment at the same condition.

-----"

Q3: For the over-lithiation of LMO, the extra lithium source originates from the pre-lithiation of Li_xTiO_2 . However, it would be better if the author could explain further where the extra lithium comes from for the lithiation of TiO_2 during the early stage of the first charge.

R3: Thanks for your comments and advice. The self-prolonging aqueous Li-ion battery (SP-ALIB) has three electrodes: the LiMn_2O_4 cathode, the TiO_2 anode, and the sacrificial Al electrode. As shown in Figure 6A, before the cycling, the battery is firstly

charged with an Al electrode and TiO₂ anode, at which time the LiMn₂O₄ electrode is not connected to the circuit. In the process, Al gets corroded while TiO₂ gets lithiated. Corresponding reaction equations are: $Al - 3e^- + xTFSI^- = Al(TFSI)_x^{3-x}$ (without HTA or PTA) / $2Al - 6e^- + 3CO_3^{2-} + 6H_2O = 2Al(OH)_3 + 3H_2CO_3$ (eg: with HTA Li₂CO₃) and $xLi^+ + TiO_2 + xe^- = Li_xTiO_2$. That's where the extra lithium comes from during the early stage of the first charge in the latter cycling.

The lithiation of TiO₂ by the sacrificial Al electrode before the formal cycling can be seen as a formation process. In this process, Li⁺ is intercalated into TiO₂ from the electrolyte, while Al(TFSI)_x^{3-x} gets into the electrode to realize the charge balance. More explanation is added in the main text to make it more clearly,

Figure R1. the program for self-prolonging ALIBs.

Main text "-----"

Before cycling, the cell is pre-charged with an Al sacrificial pre-lithiation positive electrode to get TiO₂ partially pre-lithiated (Al||TiO₂), whose pre-lithiation capacity of TiO₂ is depicted with the red short bar (Figure 6A). Subsequently, the LiMn₂O₄ electrode, instead of the sacrificial Al electrode, officially starts the cycling process (LiMn₂O₄||TiO₂)

-----"

Reply to the comments from Reviewer #2:

Comment: The manuscript is constructed based on three main logics; 1) an addition of Li_2CO_3 and Li_3PO_4 can inhibit Al corrosion, 2) a passivation process involving Al oxidation at the cathodes can serve as a lithium supplement to LiMn_2O_4 cathodes, thereby compensating for the capacity loss from HER at the anodes, 3) an introduction of a sacrificial electrode can retard Al corrosion. However, considering the significant scientific and performance concerns raised in this paper, this reviewer does not recommend its publication in Nature Communications.

Reply: Thank you for the questions and suggestions. It is conducive for us to learn about corrosion, which is helpful in improving the quality of the manuscript.

Q1: General Concept Issues:

All three of the logics are neither new nor based on correct scientific consideration.

1. The concept introduced in this paper, aiming to suppress Al corrosion with carbonate and phosphate ions, has already been reported since the 1980s (Corrosion Science (1999), 41 (9) 1743-1767; [https://doi.org/10.1016/S0010-938X\(99\)00012-8](https://doi.org/10.1016/S0010-938X(99)00012-8), CORROSION (1994), 50(3), 205–214; <https://doi.org/10.5006/1.3293512>, Corrosion and Protection of Aluminum Alloys (1986), Moscow, Publishing House Metallurgy).

R1: Thanks for providing materials for us to learn. These are precious work. Through them and some of the citations, we get a deeper understanding of the mechanism and cite them in our work. We specialize in researching how to prolong the lifetime of aqueous Li-ion batteries, among which Al corrosion is fatal. The organic salt LiTFSI is essential for existing aqueous Li-ion batteries, and it corrodes Al badly, so we mainly investigate the corrosion mechanism of TFSI and most of the references of our manuscript come from this part. The literature the reviewer mentioned above involves many methods to protect Al from the Cl^- attack, among which carbonate and phosphate ions are introduced. They are very inspiring. The principles of corrosion resistance are

often similar. Previous work laid the groundwork, but applying them to address practical issues in other electrochemical fields, such as batteries, is also a form of innovation. Thanks a lot for pointing out and providing accurate literature.

The first review, "Pitting corrosion of aluminum" is extensive, including the interaction of Cl^- with the passivation layer of Al and research of inhibitors. The text cited the 80th reference that "Citrate, tartrate, benzoate and acetate increase the pitting potential of Al but to a lesser degree than nitrate and phosphate". The conclusion clearly sets out future research directions, including pitting inhibition by inorganic compounds (chromate, nitrate, phosphate). This review is a nice learning material, inspiring researchers on the inhibitors, including phosphate.

The 80th citation of the review "Pitting corrosion of aluminum" is "The function of the passivation process in the inhibition of pitting corrosion on aluminum[J]. Corrosion Science, 1980, 20(5): 611-631". Rudd W J and Scully J C detailed work on the role of six different inhibitors in preventing Al corrosion by Cl^- , including phosphate. This work mentioned that phosphate ions act as blocking inhibitors by forming insoluble precipitates and increasing the pitting potential of Al. This article is also worth citing for us.

The second article reviewer 2 provided is "Corrosion-resistant, chromate-free talc coatings for aluminum". They provide a detailed method to build a corrosion-resistant talc coating on Al and Li_2CO_3 plays an important role. The most critical coating step is performed in 0.1M Li_2CO_3 solution, and the solution's pH is adjusted to between 11.5 and 13.5 by LiOH. Al^{3+} is also doped into the solution by immersing a sacrificial Al coupon and letting it dissolve freely. With the solution, a passivation layer of $\text{Li}_2[\text{Al}_2(\text{OH})_6]_2 \cdot \text{CO}_3 \cdot n\text{H}_2\text{O}$ is conducted on the Al surface by chemical reaction. Several conditions have to be met: 1. the native Al-oxide must be removed; 2. the reactants (Al^{3+} , Li^+ , CO_3^{2-} , and OH^-) must be present in sufficient concentration; 3. The coating must

be complete in a short time; 4. $\text{pH} > 10$. This article is instructive and some of its citations are also worth learning about the passivity of Al in alkaline solutions containing Li salts.

The work of predecessors is genuinely remarkable. However, our work is different. The electrolyte is LiTFSI solution with a pH of nearly 7, and the concentration for the battery is 10m. Under a high concentration of Li^+ , Li_2CO_3 can only solvent a little bit. Moreover, water bonds with Li^+ and enters its solvation sheath. The higher concentration, the more water bonds with Li^+ and the less free water. When the concentration of LiTFSI reaches 10m, the pH of the solution is hard to adjust by adding additives like Li_2CO_3 , as shown in Figure R2. Our primary concern to address is the Al corrosion in neutral solutions because the current battery system is based on the neutral LiTFSI solution and dependent on the relatively high concentration electrolyte to help build SEI on the anode and extend the electrochemical window ("Water-in-salt" electrolyte enables high-voltage aqueous lithium-ion chemistries[J]. *Science*, 2015, 350(6263): 938-943). Therefore, the mechanism of how Li_2CO_3 helps construct the passivation layer of Al in a neutral solution should be different from that in an alkaline solution. We did not remove the native Al-oxide. The CO_3^{2-} and OH^- are insufficient, and the pH is near 7 in 10m LiTFSI solution with saturated Li_2CO_3 , which is quite different from the abovementioned article. In our work, CO_3^{2-} tends to play a catalytic role. It combines with Al^{3+} more easily than TFSI and then $\text{Al}_2(\text{CO}_3)_3$ hydrolyzes to $\text{Al}(\text{OH})_3$ and H_2CO_3 in aqueous solution, so CO_3^{2-} returns to the solution and continues to work. We tend to believe that the passivation layer of Al is made of $\text{Al}(\text{OH})_3$ in our work instead of $\text{Li}_2[\text{Al}_2(\text{OH})_6]_2 \cdot \text{CO}_3 \cdot n\text{H}_2\text{O}$ in Li_2CO_3 and LiOH solution in the article above. We think the mechanism of the talc coating in the article above is mainly the reaction of $\text{Al}(\text{OH})_4^-$ and Al. Since Li_2CO_3 is sufficient in the solution, $\text{Li}_2[\text{Al}_2(\text{OH})_6]_2 \cdot \text{CO}_3 \cdot n\text{H}_2\text{O}$ is formed as talc coating after drying. Our mechanism is different.

Thanks for providing accurate literature, and we learned a lot. More references and introductions have been added to the main text.

Figure R2. the pH of LiTFSI solutions with or without additives.

Main text "-----"

The previous literature proposed preventing Al from chloride corrosion by constructing inorganic passivation layers^[31, 32]. Inspired by this, some insoluble aluminum compounds (Al_xM_y) such as Al(OH)₃ and AlPO₄ are selected as target passivation products due to their relatively low solubility product constant and there is literature reporting their corrosion resistance against chloride ions^[32, 33]. We apply the universal corrosion principles in the field of aqueous batteries to prevent the Al current collector from corrosion by TFSI

-----"

Q2: 1. When bare Al is exposed to the electrolyte and oxidized to Al³⁺ (Al → Al³⁺ + 3e⁻), a simultaneous reduction of the electrolyte takes place on the Al surface. 2. This corrosion reaction occurs at both the anodes and cathodes in aqueous batteries due to the low redox potential of Al/Al³⁺. Importantly, this process does not induce any pre-lithiation reactions.

R2: Thanks for your sharp thinking. It's a reasonable question considering the redox potential of Al/Al^{3+} as $-1.662V$ vs SHE. Here is the explanation.

1. It's known that Al reacts with water slowly at room temperature as $Al + H_2O \rightarrow H_2 + Al(OH)_3$ because the redox potential of Al/Al^{3+} is $-1.662V$ vs SHE ($1.378V$ vs Li/Li^+). The electrolyte for our battery is a 10m LiTFSI solution, where the electrochemical window of water is widened to $2V$ vs Li/Li^+ , as shown in Figure R3. More water is bonded with Li^+ , while free water takes up less, lowering water's reactivity. Though the redox potential of Al/Al^{3+} is still lower than the H_2 evolution reaction, we believe that Al reacts with H_2O more slowly. The native Al_2O_3 layer should also reduce the reaction speed. We believe that Al reacts with H_2O , but the slow reaction does not significantly impact the aqueous battery with the "water-in-salt" electrolyte.
2. Corrosion indeed occurs at both the cathode and anode. We focus on the corrosion issue on the $LiMn_2O_4$ cathode because of its high potential, but we are also clear that the Al current collector of the TiO_2 anode can be corroded, especially when the battery is discharged to a low voltage. However, Al corrosion on the cathode and anode causes Li intercalation in the opposite electrode. Let us take Al at $LiMn_2O_4$ cathode as Al_{LMO} and Al at TiO_2 anode as Al_{TiO} . When Al_{LMO} gets oxidized, the reactions are $Al_{LMO} - 3e^- \rightarrow Al_{LMO}^{3+}$ and $xLi - xe^- + TiO_2 \rightarrow Li_xTiO_2$. When Al_{TiO} gets oxidized, the reactions are $Al_{TiO} - 3e^- \rightarrow Al_{TiO}^{3+}$ and $xLi - xe^- + LiMn_2O_4 \rightarrow Li_{1+x}Mn_2O_4$. The results are the same that Li^+ intercalates into active material, and Al^{3+} enters the electrolyte. Corrosion on both sides has a similar pre-lithiation effect. More generally, if we assume that lithium is sufficient in electrolyte, then any electrochemical oxidation reaction other than lithium can be seen as a lithium supplement for active material, and any electrochemical reduction other than lithium can be seen as lithium loss from active material. There is evidence in the supplementary information (Figure S12) that Al oxidation causes Li intercalation;

we take it here as Figure R4. The Al-TiO₂ battery is firstly charged at 2.4V for 20 minutes and then discharged at 1C. The batteries have discharge capacity, meaning that TiO₂ is intercalated by Li⁺ when Al gets oxidized. That is the foundation of our pre-lithiation method. Going back to question 1 in this section, the slow chemical corrosion of Al by water does not have a pre-lithiation effect because no electrons transfer in the circuit. However, its electrochemical corrosion will cause pre-lithiation on the opposite electrode.

The crux of this question is whether aluminum can be electrochemically deposited in our battery system. If, on one side, Al is oxidized to Al³⁺ while on the other side, Al³⁺ deposits as Al, the pre-lithiation effect will not occur. Firstly, it is hard because the potential of either cathode or anode is higher than the potential of the redox of Al/Al³⁺ (TiO₂ gets lithiated at around 1.9V vs Li/Li⁺). Secondly, there is little Al³⁺ in the electrolyte since the pH of 10m LiTFSI is near 7, and as shown in Figure R5 (Figure 2D), the concentration of aluminum ions in the solution after corrosion is on the order of ppm.

Figure R3. The electrochemical stability window of LiTFSI-H₂O electrolytes. Cited from (From “Water-in-salt” electrolyte enables high-voltage aqueous lithium-ion chemistries[J]. *Science*, 2015, 350(6263): 938-943). Reprinted with permission from AAAS.

Figure R4. The capacity of Al oxidation in 2.4V Al-TiO₂ pouch cell. (A)(B)(C) 2.4V constant voltage charging for 20 minutes and 1C constant current discharging profiles of the first three cycles with electrolyte 10m LiTFSI, 10m LiTFSI + Li₂CO₃ or 10m LiTFSI + Li₃PO₄, respectively, with the capacity-cycle number plot inside. The area of Al is 2cm².

Figure R5. Al content in electrolytes by ICP measurement after CA experiments.

Q3: The introduction of a sacrificial electrode to retard Al corrosion was reported very early in the field of corrosion, and therefore, it has been applied commercially for a significant period of time.

“<https://xapps.xyleminc.com/Crest.Grindex/help/grindex/contents/CathodicProtection.htm>”

R3: Thanks for the comment. We apologize for the lack of clarity in our explanation of the text, which led to misunderstandings. The third sacrificial Al electrode is not used to retard the corrosion of the Al current collector. It is used to be oxidized to get active material TiO₂ pre-lithiated.

To explain it more clearly, we made Figure R6, where (A) comes from Figure S11 and (B)(C) comes from Figure 6. The batteries of Figure R6A have a low capacity of about 0.2mAh with the mass of active material less than 2mg, and they are normal batteries

with only the cathode and anode without the third sacrificial Al electrode. We aim to magnify the effects of cathode corrosion of the Al current collector in the presence of lithium intercalation at the anode; hence, we fabricated low-capacity batteries and used a 2cm² Al foil as a current collector. The mass of the Al current collector is far heavier than the active material, and its corrosion changes the electrochemical curve very much in Figure R6A (the lower plateau at the charging process is Al corrosion-TiO₂ lithiation). After the mechanism demonstration, we would like to apply it to a commercial-scale battery, so two 0.5Ah batteries are constructed, as shown in Figure R6B. The blue line is from the normal LiMn₂O₄-TiO₂ battery without sacrificial Al electrode and anti-corrosion additive, just the same condition as the battery in Figure 5A, but the electrochemical curve is quite different without the lower plateau. That is not because Al is not corroded at a big battery but because the corrosion contributes too little capacity compared to active material to show up in the curve. Therefore, to address this issue, we need to enhance corrosion to provide more capacity. However, relying solely on Al current collectors for corrosion capacity would create a paradox, as we do not want excessive corrosion of the Al current collectors. Therefore, our solution is introducing a heavy sacrificial Al electrode to be corroded in a harsher environment in the formation process before cycling to provide capacity for pre-lithiation. In cycling, anti-corrosion additives like HTA or PTA protect the Al current collector. The sacrificial electrode can be designed to be removable and reusable without compromising energy density while addressing the issue of insufficient pre-lithiation capacity.

Figure R6. Different effects of Al current collector corrosion on low-capacity and high-capacity batteries. (A) small pouch cells with a capacity of about 0.2mAh. From Figure S11. (B) 0.5Ah stacked pouch cell from Figure 6. (C) the schematic diagram of the self-prolonging ALIBs.

To avoid misunderstanding, we changed the name of the sacrificial Al electrode to the sacrificial pre-lithiation Al electrode in Figure 6 and the text. Thanks for your comment.

Q4: The Al₂O₃-destructive reaction mentioned in this manuscript ($\text{TFSI}^- + \text{Al}_2\text{O}_3 \rightarrow \text{Al}(\text{TFSI})_3 + \text{O}_2 + \text{e}^-$) is considered to be highly unlikely. This reviewer believes that the Gibbs formation free energy of Al₂O₃ formation is actually larger than that of Al(TFSI)₃.

R4: Thanks for your question. We have not conducted experimental research on how TFSI disrupts Al₂O₃. The conclusion is drawn from previous studies—the 19th reference in the text ("Inhibition of anodic corrosion of aluminum cathode current collector on recharging in lithium imide electrolytes[J]. *Electrochimica Acta*, 2000, 45(17): 2677-

2684").

In our 19th reference, Wang and coworkers proposed the mechanism for how TFSI- attacks Al₂O₃: 1. TFSI_{bulk} → TFSI_{electrode} (TFSI diffuses from bulk electrolyte to Al electrode); 2. TFSI_{electrode} → TFSI_{electrode(Al)} (TFSI absorbs onto the active spot of Al surface film); 3. TFSI_{electrode(Al)} + Al₂O₃ → [Al(TFSI)_{x^{3-x}}]_{electrode} + O₂ + e⁻ (TFSI reacts with Al₂O₃); 4. [Al(TFSI)_{x^{3-x}}]_{electrode} → [Al(TFSI)_{x^{3-x}}]_{bulk} (the complex ion diffuses from Al electrode to bulk electrolyte). This is where we get inspiration.

Our opinion about this question is based on the fact that LiTFSI corrodes Al significantly, and there is ample literature supporting this, such as the references cited in our paper, references 19-21 and 24-27. Before LiTFSI corrodes Al, it should destroy its oxidation layer, Al₂O₃. Therefore, the reaction between TFSI and Al₂O₃ is supposed to happen. The potential reaction products containing Al element are limited, including AlTFSI_{x^{3-x}}, Al(OH)₃, and Al³⁺ in aqueous solution. As we discussed above, Al³⁺ is not possible because the pH of the LiTFSI solution is near 7 as shown in Figure R2. As for Al(OH)₃ does not appear in the xps spectrum of Al corroded in 10m LiTFSI solution. Without the assistance of HTA like Li₂CO₃, forming Al(OH)₃ is quite challenging. We believe Al(OH)₃ can form on the Al surface in pure water, but the situation may differ in a highly concentrated LiTFSI solution. We speculate and attribute it to the elevated concentration of TFSI in the electrolyte, potentially resulting in a more pronounced binding of TFSI with Al³⁺ compared to OH⁻. The CO₃²⁻ carries a higher charge, making it more prone to binding with Al³⁺ than TFSI. We speculate this is a critical factor in determining the formation of Al(OH)₃. Returning to the main topic, our experimental observations confirm that Al(OH)₃ is also unlikely to be the corrosion product of the reaction of Al₂O₃ and TFSI in 10m LiTFSI solution. Therefore, the reaction that Al₂O₃ + TFSI → Al(TFSI)_{x^{3-x}} + O₂ + e⁻ is more likely to be true.

Figure R7. XPS spectra of Al foil before and after 60s Ar^+ sputtering. (A) Al foil after CA experiments in 10m LiTFSI solution. (B) Al foil after CA experiments in 10m LiTFSI solution + Li_2CO_3 . The blue dashed line is based on the Al_2O_3 peak of the pristine Al.

Q5: Insoluble compounds, such as Al_2O_3 and $\text{Al}(\text{OH})_3$, invariably form in the aqueous electrolyte on the Al surface even without salts, and help to reduce the surface area of bare Al. This is a fact that is well-established and not novel. To suppress Al corrosion significantly, the rate of precipitation reactions should outpace the rate of Al oxidation reaction. Furthermore, the precipitates should include minimal cracks and defects, and should firmly adhere to the Al surface. However, general Al compounds, including Al_2O_3 , $\text{Al}(\text{OH})_3$, and AlPO_4 , are hard to meet these criteria, resulting in limited inhibition effectiveness (Please check the above literature in General Concept Issues 1). Consequently, the addition of Li_2CO_3 and Li_3PO_4 attempted in this study does not exhibit a marked suppression effect on Al corrosion (Figure 2a and 2b; a leakage current of several $\mu\text{A cm}^{-2}$ was observed at mild potentials of 4.5 V. The leakage current should remain below $\mu\text{A cm}^{-2}$ even at high potentials

R5: Thanks for your professional comment and scientific instruction. It is reasonable and informative but may differ from our original design intent. We should present the design concept more prominently in the paper.

- 1. The introduction of anti-corrosion additives is not intended to provide absolute protection to Al but rather to extend the lifespan of aqueous lithium-ion batteries in a cost-effective and scalable manner. Our goal is to enable the use of Al current collectors in aqueous lithium-ion batteries, and according to the barrel principle, the key is to ensure that Al current collectors do not become the shortest-lived component in the aqueous lithium-ion battery. In other words, we can achieve our objective by slowing down the corrosion of Al and extending its lifespan. Even more crucial is that corrosion is typically seen as something to be avoided at all costs. However, from another positive perspective, the lithium supplementation effect brought about by oxidative corrosion can compensate for the loss of lithium in active materials during battery cycling. Therefore, we consider controlled corrosion to be a beneficial factor.*
- 2. The comment "To suppress Al corrosion significantly, the rate of precipitation reactions should outpace the rate of Al oxidation reaction" is reasonable and easy to understand. However, there is also another perspective on this question. When Al gets electrochemically corroded initially, the precipitation rate may not match the corrosion rate. Though the passivation layer may grow more slowly, the corrosion rate decreases at the area where the precipitation covers. Over time, as the passivation layer deposits to cover nearly every inch of Al's surface, the corrosion of aluminum is controlled at an acceptable rate. Figure R8 (cut from Figure 2) shows that the leakage current density decreases with time when HTA or PTA is added. After 2h, the leakage current density is about $2\mu\text{A cm}^{-2}$ with Li_2CO_3 or Li_3PO_4 (the left vertical axis is in logarithmic scale, which may make the data appear larger than it is). It is not far from the requirement below $1\mu\text{A cm}^{-2}$. Moreover, LiMn_2O_4 is the primary cathode material we focus on, and the potential of the positive electrode is below 4.5V vs Li/Li^+ most of the time during battery*

cycling. Therefore, we believe this requirement might be achievable under more lenient conditions in real battery (lower potential and less electrolyte).

3. The experiments in Figure 4 in the text have been done to demonstrate that corrosion is controllable. The corrosion curve is highly pronounced during the first charging process but virtually disappears during the second charging cycle, meaning that the anti-corrosion additives do play a role in controlling it.

Thanks for your thorough comment and discussion.

Figure R8. Chronoamperometry (CA) experiments in 10 m LiTFSI solutions.

Q6: Figure 5: Cycling performance comparisons should be conducted at a slow C-rate (e.g. 0.1 C-rate).

R6: Thanks for your advice. Corresponding experiments have been done, and the mechanism towards this question will be discussed in detail.

Firstly, all the cells in Figure 5 are designed to demonstrate that the pre-lithiation effect by Al corrosion can prolong the lifetime of aqueous batteries, as we discussed in Q3

and Figure R6. Therefore, the capacity of such cells is less than 0.2mAh, and the mass of the active material is less than 2mg. Moreover, an extra 2cm² Al foil is attached to the LiMn₂O₄ electrode to magnify the corrosion effect. All the batteries run at 3C in Figure 5. The selection of a 3C rate resulted from careful consideration, and here are our reasons.

A similar cell was constructed with 1.072mg TiO₂ and 2cm² extra Al foil. The theoretical specific capacity of TiO₂ is 168mAh/g, so the current should be 0.0179mA with a 0.1C rate. As shown in Figure R9A, the charging process is abnormal, and the battery voltage cannot increase. Typically, we set the charging cutoff voltage to 2.5V, but when the specific capacity is more than 250mAh/g, the voltage only reaches 0.6V. The small cell fails to run at 0.1C, and the reason is that the current generated by the electrochemical corrosion of Al is already sufficient to supply the current required for the TiO₂ anode 0.1C charging. We conducted an experiment in which we assembled an Al-TiO₂ battery using 2cm² Al foil as the positive electrode and subjected it to 2.4V constant voltage charging, as shown in Figure S12. Here, we present the corresponding current-time (*I-t*) curve in Figure R9B. At the end of the 20-minute 2.4V constant voltage charging, the 2cm² Al foil is still able to provide a current of 0.288mA, which means that in order to reach a voltage above 2.4V within 20 minutes, the current for constant current charging must be greater than that value. Only in this way will oxidation reactions occur beyond Al corrosion (the lithium extraction from LiMn₂O₄), providing additional current to meet the constant current charging requirements. This allows the positive electrode potential to rise, and the charging cutoff condition of 2.5V can be triggered. The cell in Figure R9 has only 1.072mg TiO₂, and the 0.1C rate equals 0.0179mA. Therefore, the 2cm² Al foil can readily provide 0.0179mA of current even when the battery voltage is only 0.6V. LiMn₂O₄ never has the opportunity to get lithium extraction because Al corrosion occurs at a potential below it, taking precedence. We chose 3C to present the mechanism in Figure 5 instead of 0.1C.

However, the cycle at 0.1C can be achieved in large-capacity batteries because the

mass loading is much higher, and the mass of Al current collector is limited. Two 0.5Ah pouch cells were constructed and cycled at 0.1C, as shown in Figure R10.

Thanks for your advice. The 0.1C cycling rate is more convincing and helps improve the quality of our manuscript.

Figure R9. (A) The first charge profiles of the charging process of a small pouch cell at 0.1C. (B) The current-time curve of the small pouch Al-TiO₂ cell in Figure S12A at the 2.4V constant voltage charging process for 20 minutes.

Figure R10. The cycle of the 0.5Ah batteries at 0.1C (A) The normal LiMn₂O₄-TiO₂

cell with 10m LiTFSI electrolyte. (B) Al||LMO||TiO₂ self-prolonging aqueous Li-ion battery with 10m LiTFSI + Li₂CO₃.

Figure R10 has been added into the Supplementary Information as Figure S15.

Q7: Ti is widely employed as a cathode current collector due to its high stability in the aqueous electrolyte even at high potentials over 5V (Journal of Solid State Electrochemistry, 26, 85-95, 2022). This reviewer could not understand why the leakage current from the Ti current collector is so high in this paper (maybe the purity issue of the Ti current collector used in this study).

R7: Thanks for your comment. The materials you provided are highly relevant and worth reading, learning from, and citing. Your speculation about why the leakage current is so high is likely correct.

We have read the paper by Walter Giurlani and his collaborators. From Figure 6B, it can be deduced that Ti exhibits a leakage current density of approximately 1.25 μ A/cm² at a potential of 4.75V in a 10m LiTFSI solution. This is indeed very small. Our Ti foil is used for coating battery electrodes. It was purchased in large quantities and is relatively inexpensive compared to laboratory-grade titanium foil, so it is indeed possible that it may not be as pure. To uncover the truth, we purchased Ti foil with a purity of 99.94% and conducted the experiments three times, as shown in Figure R11. The corrosion current density is lower than the data initially shown in Figure 2. However, it is slightly larger than the data reported in the literature but within the same order of magnitude. This might be due to factors such as instrument error, material purity, and Ti foil area measurement errors. Another reason could be that our experiment involves a 10-minute exposure to a high potential of 4.5V, followed by a 1-minute open-circuit potential holding, making the corrosion conditions harsher. This experimental design causes the corrosion current density in each segment to decrease from high current to low current, which might result in a slightly higher average current density over a short period.

Figure R11. Chronoamperometry (CA) experiments in LiTFSI with Ti current collector. (A) Figure 6B from "Electrochemical stability of steel, Ti, and Cu current collectors in water-in-salt electrolyte for green batteries and supercapacitors[J]. *Journal of Solid State Electrochemistry*, 2022, 26(1): 85-95". The blue line represents the corrosion curve of Ti at 4.75V in 10m LiTFSI. (B) CA experiments of Ti in 10m LiTFSI at 4.5V.

We replaced the data of Ti in Figure 2B and remade Figure S6 and Figure S7. Some of the statements in the paper have also been revised.

Main text "-----"

Ti is a highly corrosion-resistant current collector demonstrated in aqueous batteries^[34].

Compared with Ti, Al foils with Li_3PO_4 ($2.35 \times 10^{-6} \text{ A}/\text{cm}^2$) and Li_2CO_3 (2.71×10^{-6}

A/cm^2) are of the same order of magnitude as Ti ($5.34 \times 10^{-6} \text{ A}/\text{cm}^2$) (Figure 2B and

Figure S7).

-----"

Thanks for your detailed questions. We also appreciate the opportunity provided by the Nature Communication platform to learn from a corrosion expert. The literature you provided has been highly precise and beneficial to us. Answering your questions has deepened our understanding of the mechanisms discussed in our paper, thereby improving the quality of the article.

Reply to the comments from Reviewer #3:

This is a very interesting work on the passivation of Al current collectors and their use in long-lifetime aqueous batteries. The additives are helpful in extending the cycle time and improve the coulombic efficiency. I would like to recommend its publication. The following comments should be addressed in the revision.

Reply: We greatly appreciate your positive feedback. Your advice is practical and specific, and it has been immensely helpful in improving the quality of our manuscript.

Q1: Figure 1 is a bit complicated to follow. The authors are suggested to make it clearer and more focused

R1: Thanks for your advice. A clear and concise concept diagram can help readers understand the article's core.

Figure R12. The schematic of the HTA and PTA passivating Al mechanism in an aqueous LiTFSI electrolyte. PTA: $\text{Al}(\text{s}) - \text{e}^- + \text{X}^{\text{n-}}(\text{aq}) \rightarrow \text{Al}_x\text{X}_3(\text{s})$; HTA: $\text{Al}(\text{s}) - \text{e}^- + \text{M}^{\text{n-}} + \text{H}_2\text{O} \rightarrow \text{Al}(\text{OH})_3(\text{s}) + \text{H}_n\text{M}$.

We have updated a revised illustration version as Figure R12 to replace Figure 1.

Q2: The passivation layer is the key. The main characterizations are by XPS. The references of the XPS peaks and the analysis should be provided in detail.

R2: Thanks for your advice. We have made corresponding explanations in the manuscript.

The most critical XPS peak is Al_2O_3 because we can only determine the presence of other passivation layers by evaluating the shift relative to this peak position. However, the XPS peak position of Al_2O_3 can change with varying thicknesses, and different laboratories use different aluminum foils. Therefore, we collected XPS spectra using the pristine Al foil from our laboratory and used it as a reference, as shown in Figure 2. As for the peaks of $Al(OH)_3$ and $AlPO_4$, the $Al(OH)_3$'s peak position is 74.4eV from the reference "Characterization of Venezuelan laterites by X-ray photoelectron spectroscopy[J]. Journal of electron spectroscopy and related phenomena, 1996, 82(3): 135-143" and the $AlPO_4$'s peak position is also 74.4eV from the reference "Characterization of vanadia supported on amorphous $AlPO_4$ and its properties for oxidative dehydrogenation of propane[J]. Applied Catalysis A: General, 1994, 112(2): 187-208". Corresponding information has been added to the text.

Main text "-----"

Considering the insulating aluminum oxide in the pristine Al foil, we used their XPS as a reference (Al_2O_3 , Al spectrum: 74.81eV). Figure 2E shows that the $Al(OH)_3$ is detected at 74.27 eV for Li_2CO_3 and 74.19 eV for Li_2SiO_3 , and the $AlPO_4$ is detected at 74.37 eV for Li_3PO_4 , whose values are consistent with the literature.

-----"

Q3: In Figure 5d, the coulombic efficiencies of the first few cycles are low. Why? I do not think the charge transfer to form the passivation layer is large enough to significantly lower the coulombic efficiency.

R3: Thanks for your question. This is a question worth discussing. We will answer this question from both scientific and technological perspectives.

1. Firstly, let us discuss it from a scientific perspective. Let the Coulombic efficiency be denoted as CE, the charging specific capacity as C_{charge} , and the discharging specific capacity as $C_{discharge}$. Then CE is equal to $C_{discharge}/C_{charge}$. Due to excess positive material, we calculate specific capacity based on the negative electrode mass. TiO_2 is our anode material, and its theoretical specific is 168mAh/g, so the upper limit of the $C_{discharge}$ is typically 168 mAh/g (If we ignore other minor oxidation reactions on the negative electrode during the discharge process, such as Al corrosion at the potential of TiO_2). However, C_{charge} doesn't have such a limit. In extreme conditions, as demonstrated in our response to the sixth question of review 2#, Al corrosion can provide capacity far exceeding the theoretical capacity of TiO_2 , as shown in Figure R9. The main reaction is $Al - 3e^- \rightarrow Al^{3+}$ and $xLi^+ + xe^- + TiO_2 \rightarrow Li_xTiO_2$. When TiO_2 is fully lithiated, the main reaction becomes $Al - 3e^- \rightarrow Al^{3+}$ and $2H_2O + 2e^- \rightarrow H_2 + 2OH^-$, which could provide much larger capacity. The efficiency becomes quite low when the numerator has an upper limit and the denominator can become very large. Theoretically, the coulombic efficiency could be as low as 50% and even lower. This is closely related to the specialized battery depicted in Figure 5 that we used. The capacity is less than 0.2mAh, and the mass of active material is less than 2mg. To magnify the demonstration of the corrosion-driven pre-lithiation mechanism, we additionally attached a 2cm Al foil, which further accentuates the efficiency issue.
2. From a scientific perspective, we explained the rationality of why the efficiency could be very low. Next, we will explain the inevitability of this low efficiency from a technological perspective. The lithium intercalation potential of TiO_2 is about 1.9V (which varies with overpotential). $LiMn_2O_4$ has two voltage plateaus, approximately at 4.2V and 4.4V, with each plateau contributing roughly half of the capacity. The P/N ratio is about 2:1 by mass weight. Moreover, the cut-off voltage

of battery charging is set to 2.5V. In other words, to reach voltage cutoff, all the lithium from the 4.2V plateau of LiMn_2O_4 should be extracted, and the potential then reaches 4.4V for the battery to cut off. As we mentioned above, the P / N ratio is about 2:1, which means the capacity from the 4.2V plateau of LiMn_2O_4 is nearly enough for the lithiation of TiO_2 . However, much capacity is provided by Al corrosion before LiMn_2O_4 . This results in the fact that when TiO_2 is fully lithiated, the capacity from the 4.2V plateau is not completely depleted, and the charging process cannot be cut off. This leads to the continuation of the reaction, where the positive electrode undergoes lithium deintercalation from LiMn_2O_4 (4.2V), while the negative electrode can only provide capacity through hydrogen evolution from H_2O . The huge hydrogen evolution results in low coulombic efficiency. Then, the potential plateau of LiMn_2O_4 reaches 4.4V, and the charging process cuts off.

Q4: For Al|| LiMn_2O_4 || TiO_2 full cells in Figure 6, what are the mechanisms for capacity decay? The additives and passivation of Al current collectors improve the capacity retention by 17% over 200 cycles. A comparison of the cell impedance before and after the cycling should be given.

R4: Thanks for your question and advice. We will discuss the mechanism of capacity in detail, and the electrochemical impedance spectra before and after battery cycling have been done.

1. *As for the capacity decay of aqueous Li-ion batteries, the direct reason is Li loss from side reactions. The common side reactions are hydrogen evolution and SEI formation. Comparatively, the former is the primary reason, while the latter mainly occurs in the early stages of cycling and accounts for a relatively smaller proportion. Hydrogen evolution primarily stems from two aspects, Hydrogen evolution primarily stems from two aspects, one being electrochemical hydrogen evolution and the other being chemical hydrogen evolution. The electrolyte we used is 10m LiTFSI, which is quite a challenge for the LiMn_2O_4 - TiO_2 aqueous*

battery. As shown in Figure R3, this electrolyte's cathodic electrochemical stability window is 2V vs Li/Li⁺ but the potential of TiO₂ lithiation is about 1.9V vs Li/Li⁺. The electrochemical hydrogen evolution is inevitable and happens during every charging process, which leads to a constant Li loss. From another perspective, self-discharge is also occurring at the negative electrode continuously, the corresponding reaction being $\text{Li}_x\text{TiO}_2 + \text{H}_2\text{O} \rightarrow \text{H}_2 + \text{LiOH} + \text{TiO}_2$. The only difference is that there is no electron transfer in the circuit. Fundamentally, their principles are similar due to the low lithium intercalation potential of TiO₂. The capacity decay mechanism is attributed to the Li loss from side reactions, mainly hydrogen evolution.

2. The electrochemical impedance spectra before and after battery cycling is shown in Figure R13. The 0.5Ah pouch cells with 15 layers exhibit a complex composition in their EIS spectrum. In Figure R13B, there are more semicircles in the spectrum, which we believe may represent the contribution of the Al passivation layer Al(OH)₃ from HTA Li₂CO₃.

Figure R13. The electrochemical impedance spectra (EIS) of 0.5Ah pouch cells. (A) LMO || TiO₂ pouch cell with 10m LiTFSI. (B) Al || LMO || TiO₂ pouch cell with 10m LiTFSI + Li₂CO₃.

Thanks again for your positive feedback; it has made our article more accurate and reader-friendly.

Reply to the comments from Reviewer #4:

This paper describes a new method of passivating aluminum current collectors for use in aqueous Li-ion batteries, which has potentially significant implications for the commercial development of the technology. However, I have a concern around the lack of information provided by the authors regarding the repeatability of their measurements. No mention is made of the number of repeat tests, there are no error bars in any of the plots and no statements of measurement uncertainty are provided. This must be addressed before publication can be considered.

Reply: Thanks for your positive feedback; your advice was very meaningful, and we conducted additional experiments to enhance the credibility of our data, improving the quality of our manuscript.

The corrosion experiments in Figure 2 were repeated two more times, and we compiled some of the data to create a line graph with error bars as shown in Figure R14 and Figure R15. The repeatability of the corrosion experiments is good, and it closely matches the raw data in Figure 2. After adding HTA or PTA, there is a significant reduction in corrosion current density, and both Li_2CO_3 and Li_3PO_4 consistently perform well. The longer error bars in some of the data in Figure 5 may be attributed to several potential factors. Firstly, the Al foils used in the corrosion experiments were not stripped of their oxide layers, so the imperceptible damage or different active sites on the surface of each Al foil could have different effects on the test results. Secondly, we chose the average current density from the last segment of the CA experiment because it represents the stable performance of excellent corrosion-resistant additives after enduring the corrosion test, as seen in cases like Li_2CO_3 and Li_3PO_4 . In experimental groups with more pronounced corrosion, the uncontrolled impact of prolonged corrosion on the Al foil in the early stages led to fluctuations in the data in the final segment.

To demonstrate the repeatability of battery cycling, additional 0.5Ah Al || LMO || TiO₂ pouch cells were remade and presented in Figure R16. These three batteries had capacity retention rates of 66.9%, 70.2%, and 70.1% after 200 cycles. The repeatability is quite good, and we replace the data in Figure 6 with the data from the new battery.

Figure R14. The repeatability of the corrosion chronoamperometry experiments of Al in 1m LiTFSI solutions with additives. (A)(B) Two sets of repeated experiments as in Figure 2A. (C) The average current density of the last segment of the CA experiment in each of the three trials. (D) Statistical averaging and error analysis of the data in (C).

Figure R15. The repeatability of the corrosion chronoamperometry experiments of Al in 10m LiTFSI solutions with additives. (A)(B) Two sets of repeated experiments as in Figure 2B. (C) The average current density of the last segment of the CA experiment in each of the three trials. (D) Statistical averaging and error analysis of the data in (C).

Figure R16. The repeatability of the 0.5Ah Al || LMO || TiO₂ pouch cell with 10m LiTFSI + Li₂CO₃.

We replaced the data in the paper with the average data of three CA corrosion tests,

substituted the new battery data in Figure 6, remade Figure S6 and Figure S7, and added Figure S16.

Main text "-----"

The CA experiments in Figures 2A and 2B are repeated two times to confirm double our claims (Figure S6 and Figure S7).

Compared with Ti, Al foils with Li_3PO_4 ($2.35 \times 10^{-6} \text{ A/cm}^2$) and Li_2CO_3 ($2.71 \times 10^{-6} \text{ A/cm}^2$) are of the same order of magnitude as Ti ($5.34 \times 10^{-6} \text{ A/cm}^2$) (Figure 2B and Figure S7). Among all candidates, the HTA of Li_2CO_3 and PTA of Li_3PO_4 is much better than that of Li_2SiO_3 ($3.56 \times 10^{-5} \text{ A/cm}^2$)

Three SP-ALIBs (0.5Ah) were assembled and evaluated to validate the reliability of our design in Figure S16, and the consistency was excellent.

-----"

REVIEWER COMMENTS

Reviewer #2 (Remarks to the Author):

This reviewer believes that the paper lacks scientific novelty, such as new salts, mechanisms, or newly discovered facts using advanced technologies. Additionally, there is no significant improvement in electrochemical performance, both of which are essential to meet the high standards of the distinguished journal, Nature Communications.

Major issues

1. The author asserted that the combination of PO₄³⁻ and CO₃²⁻ and high salt concentration improved to suppress the Al corrosion in aqueous system. However, using additives to suppress Al corrosion in aqueous environment has been widely explored for most common organic and inorganic anions. Except aggressive anions such as Cl⁻, many anions show somewhat corrosion prevention effect (i.e. the PO₄³⁻ anion reported in this study (reference 30-32 in the main text and *Electrochimica Acta* 45 (2000) 1901-1910, *Acta Metallurgica Sinica*, 1997,10,1, *Corrosion Science* 68 (2013) 14-24, and many more) and many of organic anions (*Electrochimica Acta* 45 (2000) 2677-2684, *Solid State Sciences* 4 (2002) 1385-1394, *Electrochimica Acta* 49 (2004) 1483-1490, *J Solid State Electrochem* (2016) 20:507-516, any many more)). The fact that high salt concentration helps to mitigate Al corrosion in both aqueous and non-aqueous systems is also well studied (*ACS Appl. Mater. Interfaces* (2019) 11, 49, 45554-45560, *Journal of Power Sources* (2013), 231, 1, 234-238, *Energy Environ. Sci.*, (2014), 7, 416-426, and many more). Thus, this manuscript includes no dramatic or unprecedented results and represents a routine improvement which is not scientifically inspiring.

2. To demonstrate the concept, the author employed three-electrode cells with a substantial amount of electrolyte, ensuring a continuous supply of electrolyte additives (Figures 2-5). However, as the authors noted, the solubility of Li₂CO₃ and Li₃PO₄ is less than 0.05 m. Furthermore, the prevention of Al corrosion requires a large amount of PO₄³⁻ and CO₃²⁻ anions and at least 10 cycles (Figure S11), as they exhibit moderate Al corrosion prevention effects. Consequently, only a slight improvement in the electrochemical performance of the pouch cell, which contains a small amount of electrolyte, was achieved (Figure 6). This suggests that the author's approach has a limited impact on real battery systems. Besides, no postmortem analysis was conducted on the cycled pouch cell to examine the degree of corrosion of the Al current collector.

3. In the pouch cell, the author introduced an additional sacrificial Al foil. As this reviewer mentioned in the first round, the use of a sacrificial electrode to mitigate Al corrosion has been reported for quite some time and has practical applications in the electrochemical field, except for batteries. This is because the additional sacrificial foil reduces the energy density of the battery. Therefore, there is no scientific novelty in this approach.

Also, this reviewer would like to emphasize that the function of the extra Al foil does not appear to align with the authors' claims. If the original Al attached to the cathode's active materials is severely corroded and insufficient in quantity, it should result in either holes in the original Al film or the generation of a significant amount of electrically insulating corrosion product between the active materials and Al. In both scenarios, the original Al loses its contact with the active material, making it impossible to maintain capacity. Even if the Al is indeed consumed, a straightforward solution would be to use a thicker Al foil as a current collector. The authors' explanation does not seem to justify the use of a larger additional Al

foil.

Minor issues

1. page3, line54.

The author should give more explanation or reference on why the “amphoteric” nature of Al in aqueous electrolyte is related to the ease of electrochemical corrosion of Al in aqueous electrolytes.

2. page4, line82.

The role of Al oxidation (corrosion) in this study is not as a “lithium supplement” since Al cannot directly provide Li ion to the electrolyte. More appropriately, the Al is a “electron supplement” in this study.

3. page4, line84.

The reviewer could not understand the meaning of the technical term “electrochemical-chemical passivation”.

4. page9, line192.

The author should give more explanation or reference on why they think Al is interacting with CO₃²⁻.

5. page10, line215.

The term “pre-lithiate” usually means to lithiate active materials before the assembly of cells. However, in this study the “prelithiation” is happening at the same time of battery initial charging. Is the term “pre-lithate” appropriate in this context?

Reviewer #5 (Remarks to the Author):

Two types of salt additive (HTA and PTA) were studied for their capabilities of forming stable passivation layer on aluminum current collector. It is identified those additives partially replace the existing native oxide layer with Al(OH)₃ or Al-anion precipitate such as AlPO₄ and both types of layer effectively suppress pitting corrosion reaction of aluminum current collector. The extent of corrosion reaction was comparable to that of titanium current collector which is known to be highly stable under aqueous medium, which implies the possibility of the practical application of aluminum current collector for aqueous LIB. This research doesn't seem to have significant originality in terms of research idea or strategy considering the fact that aluminum corrosion issue (in relation with passivation layer) has been well-established in non-aqueous LIB system and a lot of similar passivation-research have been conducted. However, considering the meaning of enabling the application of aluminum current collector in aqueous media and the simplicity of the method (adding ~0.05m additive), the HTA and PTA additive strategy can be evaluated as having scientific importance and practical value worth sharing. Hence, I would support the publication of this work on the condition that following comments and questions are properly handled. Detailed comments are given below.

1. The authors should reexamine the reaction mechanism of HTA and PTA.

1-1. It is claimed that TFSI anion reacts with Aluminum oxide layer and then exposed Al surface reacts with HTA or PTA anion forming passivation layer. However, considering the double-layer structure of passivation layer it also seems possible that HTA or PTA reacts

with Aluminum oxide layer preferentially and form passivation layer on it, suppressing the reaction of TFSI anion.

1-2. On a related note, if the reaction between TFSI anion and Aluminum oxide layer has to occur first for the reaction between Al(s) and additive anion to occur, could the effects of HTA and PTA be insignificant or absent in salt systems other than LiTFSI?

1-3. It is claimed that HTA anion forms aluminum-anion intermediate before hydrolyzation. It is acceptable the presence of HTA is necessary for the hydrolyzation to occur according to the XPS analysis result. However, it is not clear if HTA forms aluminum-anion intermediate. Or is it not the role of the HTA anion to serve as a pH buffer in proximity to the aluminum current collector surface?

2. The over-lithiation of LMO occurs at low voltage level. Comparing the two graphs in Figure 3A, the over-lithiation strategy also has a drawback since it results in the cell's operating voltage being lowered while the capacity remains the same. In a practical battery system where energy should be provided at a constant voltage level, such design could be a serious disadvantage rather than an advantage. The authors need to present a balanced and comprehensive perspective that takes this into account. In addition, it is recommended to present comparative cycling stability data of "Al current collector, HTA or PTA additive" vs "Ti current collector" under constant high voltage system (no over-lithiation).

Reply to the comments from Reviewer #2:

Comments: This reviewer believes that the paper lacks scientific novelty, such as new salts, mechanisms, or newly discovered facts using advanced technologies. Additionally, there is no significant improvement in electrochemical performance, both of which are essential to meet the high standards of the distinguished journal, Nature Communications.

Q1: The author asserted that the combination of PO_4^{3-} and CO_3^{2-} and high salt concentration improved to suppress the Al corrosion in aqueous system. However, using additives to suppress Al corrosion in aqueous environment has been widely explored for most common organic and inorganic anions. Except aggressive anions such as Cl^- , many anions show somewhat corrosion prevention effect (i.e. the PO_4^{3-} anion reported in this study (1-3: reference 30-32 in the main text and 4: *Electrochimica Acta* 45 (2000) 1901-1910, 5: *Acta Metallurgica Sinica*, 1997,10,1, 6: *Corrosion Science* 68 (2013) 14-24, and many more) and many of organic anions (7: *Electrochimica Acta* 45 (2000) 2677-2684, 8: *Solid State Sciences* 4 (2002) 1385-1394, 9: *Electrochimica Acta* 49 (2004) 1483-1490, 10: *J Solid State Electrochem* (2016) 20:507-516, any many more)). The fact that high salt concentration helps to mitigate Al corrosion in both aqueous and non-aqueous systems is also well studied (11: *ACS Appl. Mater. Interfaces* (2019) 11, 49, 45554-45560, 12: *Journal of Power Sources* (2013), 231, 1, 234-238, 13: *Energy Environ. Sci.*, (2014), 7, 416-426, and many more). Thus, this manuscript includes no dramatic or unprecedented results and represents a routine improvement which is not scientifically inspiring.

R1: Thank you for the reviewer's comment. We made further efforts to claim our innovation based on the reviewer's concern.

Firstly, we agree with the reviewer that corrosion and passivation of Al have been extensively studied, and some protective ions have been reported. However, the mechanisms through which these ions form a passivation layer to protect Al have not

been elucidated as clearly and precisely as presented in this manuscript. Understanding the corrosion mechanisms and applying this knowledge in batteries are pronounced, particularly in aqueous batteries. We believe the transdisciplinary application and knowledge migration represent a form of innovation.

Second, thank you for listing so much helpful literature. Some are cited in the main text, and a more accurate description is added.

Main text

"----- Different from the non-aqueous electrolytes where Al can be passivated through adding effective salts like LiPF₆, LiBF₄, LiDFOB or optimizing solvents with lower dielectric constant^{11,19-29}, it is much more challenging to passivate the Al metal foil in the aqueous solution because of the incompatibility of the salts and water. On the one hand, the solvent water has a relatively high dielectric constant (78.4F/m at 25°C), promoting the diffusion of the corrosion product and accelerating Al corrosion^{19,20}; -----"

Additionally, we numbered the literature the reviewer mentioned for the convincement discussed them as follows:

Inorganic ion: literature 1-6.

- 1. (Lv T, Suo L. Water-in-salt widens the electrochemical stability window: Thermodynamic and kinetic factors[J]. Current Opinion in Electrochemistry, 2021, 29: 100818.)*

This is the 30th reference in the main text of the first round. The thermodynamic and kinetic factors of the electrochemical stability window of the "Water-in-salt" aqueous electrolyte are mainly discussed. Few corrosion issues are involved.

- 2. (Szkłarska-Smiałowska Z. Pitting corrosion of aluminum[J]. Corrosion science, 1999, 41(9): 1743-1767.)*

This is the 31st reference in the main text of the first round of reversion. We have discussed this review in the first round of revision. In this review, PO₄³⁺ is mentioned in this sentence: " Citrate, tartrate, benzoate and acetate increase the pitting

potential of Al but to a lesser degree than nitrate and phosphate [80]”.

We checked the 80th reference to find out the effect of PO_4^{3-} . (Rudd W J, Scully J C. The function of the repassivation process in the inhibition of pitting corrosion on aluminium[J]. Corrosion Science, 1980, 20(5): 611-631.). This literature provides experimental determination of pitting corrosion potentials for Al in Cl⁻ solutions, comparing the passivation effects of various anions, including PO_4^{3-} . Indeed, PO_4^{3-} has been demonstrated to enhance the pitting corrosion potential of Al, albeit within a Cl⁻ system, which diverges from the aqueous lithium-ion electrolyte system employed in this study. Our manuscript applies fundamental corrosion principles to address practical challenges arising from TFSI in aqueous battery systems, highlighting substantial distinctions.

3. *(Rudd W J, Scully J C. The function of the repassivation process in the inhibition of pitting corrosion on aluminium[J]. Corrosion Science, 1980, 20(5): 611-631.).*

This is the 80th reference of the 2nd literature we discussed above.

4. *(Lee W J, Pyun S I. Effects of sulphate ion additives on the pitting corrosion of pure aluminium in 0.01 M NaCl solution[J]. Electrochimica acta, 2000, 45(12): 1901-1910.)*

This article primarily investigates the influence of SO_4^{2-} on Al corrosion in a 0.01M NaCl solution. From the perspective of our manuscript, $\text{Al}_2(\text{SO}_4)_3$ does not precipitate and thus is not considered one of PTA. SO_4^{2-} is a strong acid anion and does not undergo dual-hydrolysis with Al^{3+} . Therefore, considering this, SO_4^{2-} is not considered a candidate for HTA. Hence, this literature is not highly relevant to the framework of our manuscript.

5. *(Acta Metallurgica Sinica, 1997,10,1).*

Sorry, we cannot find the paper due to missing some information.

6. *(Abd El Aal E E, Abd El Wanees S, Farouk A, et al. Factors affecting the corrosion behaviour of aluminium in acid solutions. II. Inorganic additives as corrosion inhibitors for Al in HCl solutions[J]. Corrosion Science, 2013, 68: 14-24.)*

This literature aims to discover the role of CrO_4^{2-} , WO_4^{2-} , and HPO_4^{2-} in the kinetics of the dissolution of the Al_2O_3 film in the HCl solution. The key conclusion is that

the dissolution rate of Al₂O₃ decreases with an increase in the concentration of additives. The trend depends on the additive type, and the inhibition efficiency decreases in the following order: CrO₄²⁻ > WO₄²⁻ > HPO₄²⁻.

The literature above mainly focuses on the dissolution of Al₂O₃ in an HCl solution. However, as we know, the HCl solution is unsuitable for aqueous Li-ion batteries, and our batteries operate in a neutral environment rather than under acidic conditions. As mentioned above, the concentration of additives is essential for the anti-corrosion effect, but the anti-corrosion mechanism introduced in our manuscript demonstrates favorable outcomes with only trace amounts of HTA or PTA. Our primary focus lies in the corrosion issues of Al by TFSI under the operating conditions of practical aqueous battery systems. We believe that our article diverges significantly regarding both focal points and mechanisms from the literature above.

Summary for inorganic corrosion inhibitors.

The literature on inorganic corrosion inhibitors mentioned above primarily focuses on Al corrosion induced by Cl⁻. However, Cl⁻-containing electrolytes do not apply to our aqueous lithium-ion battery system due to issues such as electrochemical stability window and compatibility. Therefore, the outlined research does not address the corrosion challenges posed by TFSI in our practical applications. Unlike these studies, our paper presents a specific mechanism of Al passivation in a non-chloride system involving both HTA and PTA. We believe that the mechanism proposed in our work provides a clearer understanding of corrosion in aqueous battery systems, particularly in addressing the challenges posed by TFSI corrosion.

Organic salts: literature 7-10:

- 7. Wang X, Yasukawa E, Mori S. Inhibition of anodic corrosion of aluminum cathode current collector on recharging in lithium imide electrolytes[J]. Electrochimica Acta, 2000, 45(17): 2677-2684.*

It is the 19th reference of our manuscript. We have learned a lot from this literature.

We have referenced its conclusions to illustrate the mechanisms through which TFSI attacks and disrupts the Al₂O₃ and discussed it with you in the first round of revision. Another crucial finding in this article is the proposal to employ organic additives such as PF₆⁻ to inhibit TFSI-induced corrosion of Al. The inhibition of Al corrosion by PF₆⁻ is attributed to forming a fluorine-containing passivation layer through reaction. This conclusion has also been reported in various other literature sources. In line 53, we mentioned it with the primary intention of conveying that it is relatively easier in organic systems to control Al corrosion by adjusting the salt composition, such as introducing PF₆⁻ or BF₄⁻, to inhibit corrosion by forming a fluorine-containing layer. Additionally, modulating solvent components is also helpful. However, in aqueous electrolyte environments, PF₆⁻ and BF₄⁻ are unstable, and water as the solvent cannot be replaced. Therefore, these effective methods cannot be applied in aqueous battery systems, highlighting the challenges in inhibiting corrosion in such environments.

More explanation will be added in the introduction.

8. *(Nakajima T, Mori M, Gupta V, et al. Effect of fluoride additives on the corrosion of aluminum for lithium ion batteries[J]. Solid State Sciences, 2002, 4(11-12): 1385-1394.)*

This literature investigates the effect of fluoride additives (LiBF₄, LiPF₆, LiAsF₆, LiSbF₆) on the corrosion of Al in non-aqueous electrolytes. The ranking of corrosion resistance performance from best to worst is LiBF₄ > LiPF₆ > LiAsF₆ > LiSbF₆.

As we mentioned in the last paragraph, anti-corrosion organic salts like LiBF₄ and LiPF₆ cannot remain stable in aqueous electrolytes. The anti-corrosion mechanism is also quite different from ours.

9. *(Song S W, Richardson T J, Zhuang G V, et al. Effect on aluminum corrosion of LiBF₄ addition into lithium imide electrolyte; a study using the EQCM[J]. Electrochimica acta, 2004, 49(9-10): 1483-1490)*

This literature investigated the Al anti-corrosion effect of LiBF₄ in 1M LiTFSI/

EC+DMC electrolyte and determined the corrosion product and passivation layer through EQCM and FT-IR.

Protecting Al in non-aqueous electrolytes is more manageable, and some organic salts have an excellent anti-corrosion effect. However, it's not useable for aqueous electrolytes because of the instability of the organic salts. The mechanism of how LiBF_4 or LiPF_6 passivate Al is well developed. However, substantial distinctions exist between these mechanisms and ours of HTA/PTA in aqueous battery systems.

10. (Yan G, Li X, Wang Z, et al. Lithium difluoro (oxalato) borate as an additive to suppress the aluminum corrosion in lithium bis (fluorosulfony) imide-based nonaqueous carbonate electrolyte[J]. *Journal of Solid State Electrochemistry*, 2016, 20: 507-516.)

This literature demonstrated the anti-corrosion effect of LiDFOB in non-aqueous LiFSI electrolytes. During the charge process, the LiDFOB oxides at the Al's surface form a passivating film by producing polymers.

Firstly, LiDFOB is unstable with water, making it incompatible with aqueous Li-ion batteries. Secondly, the mechanism by which LiDFOB passivates Al involves oxidation during the charging phase, forming a polymer passivation film on the Al surface. Such mechanism is quite different from ours of HTA and PTA and the passivation layer formed by LiDFOB includes organic components, whereas the passivation layers formed by HTA and PTA additives are inorganic, exhibiting distinct compositional differences.

Summary for this section: organic anti-corrosion salts.

For non-aqueous electrolytes, Al can be passivated by adjusting the types of salts and solvents. LiPF_6 , LiBF_4 , and LiDFOB all have anti-corrosion effects on Al current collector. LiPF_6 and LiBF_4 can form a fluorine-contained layer on Al surface, while LiDFOB can be oxidized to form a passivating film by producing polymers. However, these salts are not incompatible with aqueous electrolytes. Thus, addressing the corrosion problem brought by TFSI in practical aqueous Li-ion

batteries is not helpful. Moreover, the mechanism and the passivation layer components above are quite different from the HTA and PTA proposed by our manuscript. From this perspective, the innovation of passivating Al current collectors with HTA and PTA in aqueous Li-ion batteries seems to be independent of the literature focusing on non-aqueous electrolytes.

High salt concentration mitigates Al corrosion: literature 11-13

11. (Ko S, Yamada Y, Yamada A. Formation of a solid electrolyte interphase in hydrate-melt electrolytes[J]. ACS applied materials & interfaces, 2019, 11(49): 45554-45560)

This article is not related to Al corrosion. It primarily discusses the mechanism of inhibiting hydrogen evolution from water at the negative electrode, where the passivating layer refers to anion-derived solid electrolyte interphase (SEI) layer. In this context, passivation refers to mitigating water decomposition rather than inhibiting Al corrosion. Additionally, the reactions involved are reduction reactions, not oxidation reactions, making them unrelated to Al corrosion. High-concentration electrolyte lowers the anion's LUMO energy level and helps the formation of the SEI layer.

12. (Matsumoto K, Inoue K, Nakahara K, et al. Suppression of aluminum corrosion by using high concentration LiTFSI electrolyte[J]. Journal of power sources, 2013, 231: 234-238.)

This article is our 23rd reference in the main text. It investigates the corrosion behavior of Al in solutions with varying concentrations of LiTFSI/EC: DEC. Its main conclusion is that higher concentration helps mitigate Al corrosion because at higher LiTFSI concentrations, it is easier to connect the Li⁺ and F⁻ generated from the oxidation of TFSI and the LiF passivation layer protects Al.

Higher concentration indeed helps mitigate Al corrosion. However, in our manuscript, a 10m LiTFSI aqueous solution is used as the electrolyte, and we did not regulate Al corrosion through varying electrolyte concentrations. The lithium salt concentration is consistent between the experimental and control groups, ensuring that the concentration's impact on corrosion remains constant. The fact is that 10m LiTFSI corrodes badly, while 10m LiTFSI with HTA or PTA performs much better. From this perspective, we believe that articles studying how concentration affects Al corrosion do not impede the novelty of our manuscript.

13. (McOwen D W, Seo D M, Borodin O, et al. Concentrated electrolytes: decrypting electrolyte properties and reassessing Al corrosion mechanisms[J]. *Energy & Environmental Science*, 2014, 7(1): 416-426.)

This article is our 22nd reference in the main text. It provides a comprehensive explanation regarding why high salt concentrations in the solvent can effectively inhibit corrosion: “For the highly concentrated solvent–LiTFSI electrolytes, it is expected that the solubility of Al–TFSI complexes will be much lower (than for dilute electrolytes) as the solvent molecules and anions are already extensively coordinated. The high TFSI⁻ concentration at the electrode–electrolyte interface (similar to the interface with ionic liquids) will also serve as a barrier that hinders the access of solvent molecules to the electrode and the loss of material (dissolution) from the electrode surface. In addition, for the concentrated electrolytes, the absence of uncoordinated solvent, relatively low amount of solvent present in general and extensive coordination of both the solvent molecules and anions to Li⁺ cations (which increases their oxidative stability) further combine to inhibit the oxidation reactions at the Al electrode surface”.

As mentioned in the last paragraph, the lithium salt concentration is consistent between the experimental and control groups in our work. Therefore, how concentration affects Al corrosion is irrelevant to our manuscript.

Summary for this section: high concentration mitigates Al corrosion.

Higher concentration typically means lower solubility of Al-TFSI complexes, thus hindering the departure of corrosion products from the surface of Al. This is a concise explanation of how high-concentration electrolytes inhibit corrosion. In our manuscript, corrosion is regulated by HTA and PTA rather than the varying electrolyte concentrations. The mechanism of how concentrated electrolytes mitigate Al corrosion differs from HTA or PTA. Therefore, it should not impede the novelty of our work.

Q2:

2-1: To demonstrate the concept, the author employed three-electrode cells with a substantial amount of electrolyte, ensuring a continuous supply of electrolyte additives (Figures 2-5). However, as the authors noted, the solubility of Li_2CO_3 and Li_3PO_4 is less than 0.05 m. Furthermore, the prevention of Al corrosion requires a large amount of PO_4^{3-} and CO_3^{2-} anions and at least 10 cycles (Figure S11), as they exhibit moderate Al corrosion prevention effects. Consequently, only a slight improvement in the electrochemical performance of the pouch cell, which contains a small amount of electrolyte, was achieved (Figure 6). This suggests that the author's approach has a limited impact on real battery systems.

R2: Thank you for your comment. We discussed these questions in detail and added a more detailed description in the main text.

R2-1: Firstly, we think HTA might not encounter the issue of insufficient quantity for passivating Al. In the first round of our revision, our response to your Q1 mentioned that: " CO_3^{2-} tends to play a catalytic role that combines with Al^{3+} more easily than TFSI to hydrolyzes to $\text{Al}(\text{OH})_3$ and H_2CO_3 in aqueous solution, so CO_3^{2-} returns to the solution and continues to work." Corresponding equations are: $\text{Al}^{3+}(\text{aq}) + \text{CO}_3^{2-}(\text{aq}) +$

$H_2O \rightarrow Al(OH)_3 + H_2CO_3$. CO_3^{2-} return to the solution and continue to play a role. Therefore, from this view, HTA would not have a limitation on quantity.

As for PTA like Li_3PO_4 , PTA is indeed consumed during the passivation process. However, after a rough estimation, we find that the amount of PTA we used is sufficient to passivate Al. Suppose we assume that the thickness of the $AlPO_4$ layer is 20 nm, based on the results from XPS in Figure 2. This assumption is derived from the fact that after Ar^+ etching for 60 seconds, the peak associated with $AlPO_4$ did not disappear. Calculating based on an etching depth of approximately 10nm after 60 seconds, the estimated thickness is around 20 nm. The density of $AlPO_4$ is 2.57 g/cm^3 , so the mass of the $AlPO_4$ on 1 cm^2 passivated Al foil is $5.14 \times 10^{-6} \text{ g}$. Take our 0.5Ah pouch cell for example, we have 8 positive electrodes measuring $4.5 \times 5.8 \text{ cm}$ each. The area is 208.8 cm^2 (417.6 cm^2 for double sides), and if $AlPO_4$ is to cover the entire surface area, then $2.14 \times 10^{-3} \text{ g } AlPO_4$ is required, which is $1.76 \times 10^{-5} \text{ mol}$ considering the molar mass of $AlPO_4$ is 122. So, the electrolyte has to contain at least $1.76 \times 10^{-5} \text{ mol } Li_3PO_4$ to guarantee the passivation. The molality of 10m LiTFSI is approximately equivalent to the molarity of 4M LiTFSI. So if the concentration of Li_3PO_4 is 0.05m (0.02M), it requires $1.76 \times 10^{-5} \text{ mol} / 0.02 \text{ (mol/L)} = 0.88 \text{ mL}$ electrolyte. The above calculations demonstrate that if the concentration of Li_3PO_4 is 0.05m, then achieving the complete passivation of the positive electrode current collector Al foil, covering it with a 20 nm $AlPO_4$ layer for a 0.5 Ah aqueous pouch cell, would require only 0.88 mL of electrolyte. The density of 10m LiTFSI is about 1.55 g/cm^3 , so 0.88 mL means 1.36 g. In other words, when the electrolyte usage reaches 2.72 g/Ah, the amount of Li_3PO_4 contained therein is already sufficient. This electrolyte usage is significantly below the commercial standard of 4.5 g/Ah. The positive electrode current collector is double-coated with $LiMn_2O_4$, and the actual area in contact with the electrolyte and undergoing passivation is much smaller than the total area of 417.6 cm^2 . Moreover, the estimated thickness of the 20nm passivation layer is likely an overestimate. For typical electrolyte usage, a

0.05m concentration of Li_3PO_4 already exceeds the required passivation concentration by a significant margin.

As for Figure S11 (now Figure S12 after revision), we apologize for not providing more detailed explanations and clearer annotations in the Supplementary Information to help readers better understand the meaning of this figure, leading to misunderstandings. It does not imply that passivation requires 10 cycles but instead aims to illustrate the difference between the lithium extraction platform for over-lithiated LiMn_2O_4 and the Al oxidation platform. The conclusion is that the passivation is almost done in the first cycle from Figure 4, and the oxidation of Al in the first charging process leads to the over-lithiation of LiMn_2O_4 . Starting from the second cycle, the platform around 1.2V is essentially the reversible lithium extraction platform of over-lithiated LiMn_2O_4 , rather than the platform related to Al corrosion. The Al oxidation plateau in the first cycle is close to the plateau of the over-lithiation LiMn_2O_4 , so we especially distinguish it in Figure S12. Upon closer inspection, it can be observed that the Al corrosion voltage platform during the first cycle is more inclined when HTA or PTA is added, corresponding to the increasing difficulty in corrosion due to continuous Al passivation. In contrast, the lithium extraction platform of LiMn_2O_4 during over-lithiation appears much flatter. Figure 4 and Figure S12 are Figure R1 and Figure R2. In the experiments depicted in Figure R1, we continuously charged one LiMn_2O_4 electrode against two TiO_2 negative electrodes, preventing the occurrence of LiMn_2O_4 over-lithiation. Therefore, the 1.2V platform could only have originated from Al corrosion. However, this corrosion platform disappeared during the second charging process, indicating that the passivation process was completed mainly after the first charge. Hence, the additional charging platform in the 10th cycle in Figure R2 is attributed to the over-lithiation of LiMn_2O_4 . Therefore, it's a misunderstanding that preventing Al corrosion requires a large amount of PO_4^{3-} and CO_3^{2-} anions and at least 10 cycles (Figure S12), as they exhibit moderate Al corrosion prevention effects.

For the electrochemical performance of 0.5Ah batteries, simply adding some Li_2CO_3 to

the electrolyte and incorporating an additional step during formation can lead to a notable 20.6% improvement in the capacity retention over 200 cycles. This is highly beneficial for the practical application of aqueous batteries.

R2-2: Thanks for your advice. To compare the corrosion of Al current collector after battery cycling, batteries with 10m LiTFSI and 10m LiTFSI + Li₂CO₃ were constructed, cycling at 1C for 100 cycles or at 3C for 200 cycles. Photos of SEM are shown as Figure R3 below.

It is evident that adding HTA significantly reduces the number of corrosion pits after cycling and small corrosion sites are challenging to connect into larger areas. In contrast, with a pure 10m electrolyte, the corrosion area is substantial, and corrosion pits readily merge into more significant regions. This validates the corrosion resistance performance of our additive.

Figure R1. Clarify the duration of Al oxidation corrosion for lithium compensation.
 (A) the schematic diagram of the devices with the double TiO_2 anodes. (B) consecutive charging processes of cells with one LiMn_2O_4 cathode and two TiO_2 anodes in 10m $\text{LiTFSI} + \text{Li}_2\text{CO}_3$ or Li_3PO_4 .

Figure R2. The potential difference between Al passivation and Li extraction from over-lithiated LiMn_2O_4 . (A) 10m LiTFSI as electrolyte. (B) (C) 10m LiTFSI with Li_2CO_3 or Li_3PO_4 as electrolyte.

Figure R3. SEM photos of Al current collector after cycling. (A) 10m LiTFSI, battery underwent 100 cycles at 1C. (B) 10m LiTFSI, battery underwent 200 cycles at 3C. (C) 10m LiTFSI with Li₂CO₃, battery underwent 100 cycles at 1C. (D) 10m LiTFSI with Li₂CO₃, battery underwent 200 cycles at 3C.

We added Figure R3 as Figure S6 in the Supplementary Information and supplied more description about the discussion quantity of HTA or PTA.

Main text

"----- Moreover, in Supplementary Information Note 2, we discussed the influence of the amounts of HTA and PTA on the corrosion-resistance. -----"

Supplementary Information

"----- This section discusses whether the HTA or PTA less than 0.05m is sufficient in the electrolyte to passivate the Al current collector.

We conclude that HTA less than 0.05m is sufficient for passivating Al. Because the CO_3^{2-} tends to play a catalytic role which combines with Al^{3+} more easily than TFSI⁻ and then hydrolyzes to $\text{Al}(\text{OH})_3$ and H_2CO_3 in aqueous solution, so CO_3^{2-} returns to the solution and continues to work.” Corresponding equations are: $\text{Al}^{3+}(\text{aq}) + \text{CO}_3^{2-}(\text{aq}) + \text{H}_2\text{O} \rightarrow \text{Al}(\text{OH})_3 + \text{H}_2\text{CO}_3$. Therefore, HTA should not be limited by the current quantity.

As for PTA like Li_3PO_4 , PTA is indeed consumed during the passivation process. However, after a rough calculation, we find that the the amount of PTA we used is sufficient to passivate Al. Assume that the thickness of the AlPO_4 layer is 20 nm, based on the results from XPS in Figure 2. This assumption is derived from the fact that after Ar^+ etching for 60 seconds, the peak associated with AlPO_4 did not disappear. Calculating based on an etching depth of approximately 10nm after 60 seconds, the estimated thickness is around 20 nm. The density of AlPO_4 is 2.57 g/cm^3 , so the mass of the AlPO_4 on 1 cm^2 passivated Al foil is $5.14 \times 10^{-6} \text{ g}$. Take our 0.5Ah pouch cell for example, we have 8 positive electrodes measuring $4.5 \times 5.8 \text{ cm}$ each. The area is 208.8 cm^2 (417.6 cm^2 for double sides), and if AlPO_4 is to cover the entire surface area, then $2.14 \times 10^{-3} \text{ g}$ AlPO_4 is required, which is $1.76 \times 10^{-5} \text{ mol}$ considering the molar mass of AlPO_4 is 122. So, the electrolyte has to contain at least $1.76 \times 10^{-5} \text{ mol}$ Li_3PO_4 to guarantee the passivation. The molality of 10m of LiTFSI is approximately equivalent to the molarity of 4M LiTFSI. If the concentration of Li_3PO_4 is 0.05m (0.02M), it requires $1.76 \times 10^{-5} \text{ mol} / 0.02 \text{ (mol/L)} = 0.88 \text{ mL}$ electrolyte. The above calculations demonstrate that if the concentration of Li_3PO_4 is 0.05m, then achieving the complete passivation of the positive electrode current collector Al foil, covering it with a 20 nm AlPO_4 layer for a 0.5 Ah aqueous pouch cell, would require only 0.88 mL of electrolyte. The density of 10m LiTFSI is about 1.55 g/cm^3 , so 0.88 mL means 1.36 g. In other words, when the electrolyte usage reaches 2.72 g/Ah, the amount of Li_3PO_4 contained therein is already sufficient. This electrolyte usage is significantly below the commercial standard of 4.5 g/Ah. The positive electrode current collector is double-coated with LiMn_2O_4 , and the actual area in contact with the electrolyte and undergoing passivation is much smaller than the total area of 417.6 cm^2 . Moreover, the estimated

thickness of the 20nm passivation layer is likely an overestimate. For typical electrolyte usage, a 0.05m concentration of Li_3PO_4 already exceeds the required passivation concentration by a significant margin.

-----"

----- SEM photos of Al current collectors after battery cycling are shown in Figure S6. -----"

Q3:

3-1: In the pouch cell, the author introduced an additional sacrificial Al foil. As this reviewer mentioned in the first round, the use of a sacrificial electrode to mitigate Al corrosion has been reported for quite some time and has practical applications in the electrochemical field, except for batteries. This is because the additional sacrificial foil reduces the energy density of the battery. Therefore, there is no scientific novelty in this approach.

R3: Thanks for your comment. We will add additional explanations in the main text to elucidate the impact of the additional Al electrode on the battery's energy density. Furthermore, we will offer a more detailed exposition of the design thinking behind the additional Al electrode, illustrating how it can protect the Al current collectors during battery cycling and facilitate capacity compensation from the additional Al electrode.

R3-1: In modern commercial batteries, Al current collectors typically account for approximately 3% of the total mass (Choudhury R, Wild J, Yang Y. Engineering current collectors for batteries with high specific energy[J]. Joule, 2021, 5(6): 1301-1305.). The sacrificial Al electrode we utilize has a mass equivalent to that of the positive current collector. This results in a less than 5% decrease in energy density, which is not substantial. If a larger surface area Al mesh is employed, its mass contribution would be even less. The fundamental reason lies in the exceptionally high theoretical specific capacity of Al, reaching 2980 mAh/g, compared to the theoretical specific capacity of the TiO_2 negative electrode at 168 mAh/g. Hence, adopting minimal sacrificial Al

electrodes theoretically provides an extra electron supplement.

3-2: Also, this reviewer would like to emphasize that the function of the extra Al foil does not appear to align with the authors' claims. If the original Al attached to the cathode's active materials is severely corroded and insufficient in quantity, it should result in either holes in the original Al film or the generation of a significant amount of electrically insulating corrosion product between the active materials and Al. In both scenarios, the original Al loses its contact with the active material, making it impossible to maintain capacity. Even if the Al is indeed consumed, a straightforward solution would be to use a thicker Al foil as a current collector. The authors' explanation does not seem to justify the use of a larger additional Al foil.

R3-2:

Our goal is to protect the Al current collector during battery cycling and utilize the additional capacity provided by the oxidation of the sacrificial Al electrode to extend the lifespan of the aqueous battery. Balancing the protection of the Al current collector while relying on the corrosion of the sacrificial Al electrode to provide capacity may seem contradictory, but our design allows us to achieve this objective. The following will provide a detailed explanation.

If the Al current collector is severely corroded, the active materials can detach, significantly impacting the electrochemical performance of the battery. Therefore, our primary objective is to protect the Al current collector, and this is the underlying purpose behind the design of HTA and PTA. After adding HTA or PTA, the Al current collector gets passivated, but it also means that there is minimal electron supplement from the oxidation of the Al current collector. Therefore, to facilitate more electron supplements, we chose to add the sacrificial Al electrode. During formation, the sacrificial Al electrode undergoes continuous oxidation under prolonged constant-voltage charging at elevated temperatures and higher potential. In such harsh conditions, even the passivation layers formed by HTA and PTA cannot fully protect the sacrificial Al electrode from corrosion. As the Al current collectors at the positive and sacrificial Al electrodes are independent, the Al current collector on the LiMn₂O₄ side

is not connected to the circuit during the formation and pre-lithiation stages. Therefore, Al current collector does not undergo electrochemical reactions in the harsh environment described above. This approach allows us to achieve pre-lithiation of the negative electrode during the formation stage using the sacrificial Al electrode while protecting the Al current collector at the positive electrode from corrosion under normal operating conditions through HTA and PTA.

The core design avoids corrosion of the current collector but uses the corrosion of the sacrificial electrode. To achieve this goal, we differentiated the operating conditions of the Al current collector and the sacrificial electrode by employing a three-electrode cell. Our Al current collector gets protected from HTA and PTA and works well during the battery's lifespan, while the sacrificial Al electrode gets oxidized in the harsh formation process to make TiO₂ pre-lithiated. We used an additional sacrificial Al electrode instead of a thicker Al current collector.

Main text

"----- It should be noted that introducing the sacrificial Al electrode would lower the energy density. However, considering the sacrificial Al electrode has a high theoretical capacity of 2980 mAh/g, it is acceptable that the overall estimated energy density would be decreased by less than 5% -----"

"----- To simultaneously achieve the protection of the Al current collector and utilize Al corrosion for electron supplement, we have to employ an additional sacrificial Al electrode and differentiate the operating conditions of the Al current collector and the sacrificial electrode. In such cases, the Al current collector is protected from HTA and PTA and works well during the battery lifespan, while the sacrificial Al electrode gets oxidized in harsher conditions to make TiO₂ pre-lithiated. -----"

Q4: page3, line54. The author should give more explanation or reference on why the “amphoteric” nature of Al in aqueous electrolytes is related to the ease of

electrochemical corrosion of Al in aqueous electrolytes.

R4: Thanks for pointing out our mistake. The word “amphoteric” here may cause misunderstanding. Therefore, we removed it and expressed our ideas with more precision.

We intended to convey that Al corrosion can be suppressed in organic electrolytes by adding LiPF₆ and LiBF₄ or adjusting the solvent composition, but achieving similar passivation in aqueous electrolytes is challenging. The term 'ampholyte' should not have been emphasized, and removing it would make the article express the intended meaning more accurately, avoiding confusion.

Main text

----- Different from the non-aqueous electrolytes where Al can be passivated through adding effective salts like LiPF₆, LiBF₄, LiDFOB or optimizing solvents with lower dielectric constant^{11,19-29}, it is much more challenging to passivate the Al metal foil in the aqueous solution because of the incompatibility of the salts and water. On the one hand, the solvent water has a relatively high dielectric constant (78.4F/m at 25°C), promoting the diffusion of the corrosion product and accelerating Al corrosion^{19,20}-----

-----"

Q5: page4, line82. The role of Al oxidation (corrosion) in this study is not as a “lithium supplement” since Al cannot directly provide Li ion to the electrolyte. More appropriately, the Al is a “electron supplement” in this study.

R5: Thank you for your advice. "Electron supplement" is a more scientifically accurate description, and we will modify it accordingly.

The process involves the oxidation of Al at the positive electrode, providing electrons. Al³⁺ enters the electrolyte, while Li⁺ from the electrolyte is intercalated in the negative electrode. The additional source of lithium in the active material comes from the electrolyte, while the Al foil is the extra electron supplement. We described the source

of lithium in response to Reviewer 1's question 3 in the previous round, and we will describe it specifically in the manuscript to make it clearer.

Main text

"----- Basically, the oxidation of Al is an electron supplement process, accompanying with the additional lithium source comes from the electrolyte as lithium supplement. Therefore, "electron supplement" is used instead of "lithium supplement" in this context.

-----"

"----- Overall, Al is oxidized into Al ion dissolving in the electrolyte, at the same time, Li-ion from the electrolyte is intercalated in the active material. The additional source of lithium comes from the electrolyte, while the oxidation of Al foil is the extra electron supplement. Thus, it is concluded that the oxidation of the Al current collector at the cathode side can play a role as a pre-lithiation additive to prolong the life of batteries. -

-----"

Q6: page4, line84. The reviewer could not understand the meaning of the technical term "electrochemical-chemical passivation".

R6: Thank you very much for your comment. More descriptions are added to make it more straightforward.

The passivation process has two steps. First, Al gets electrochemically oxidized, and then Al^{3+} combines with anions from HTA or PTA in the electrolyte to form $Al(OH)_3$ or $AlPO_4$. The former process is an electrochemical reaction while the latter process is a chemical reaction. That's why we call it electrochemical-chemical passivation.

Main text

"----- First, electrochemical corrosion occurs, followed by chemical passivation.

Thus, we refer to it as electrochemical-chemical passivation.-----"

Q7: page9, line192. The author should give more explanation or reference on why they think Al is interacting with CO_3^{2-} .

R7: Thank you very much for your comment. More explanation is added in the main text.

Initially, we used $\text{Al}_2(\text{CO}_3)_3$ as an intermediate state to help readers better understand the reaction scenario we wanted to portray, similar to PTA. The core of the passivation reaction is that the anions of the additive bind preferentially to Al^{3+} over TFSI anions. However, the existence of Al_2O_3 has not been accurately confirmed until 2023 when Lkhamsuren Bayarjargal synthesized $\text{Al}_2(\text{CO}_3)_3$ under high CO_2 pressure of 24-28 GPa (Anhydrous Aluminum Carbonates and Isostructural Compounds[J]. Inorganic Chemistry, 2023, 62(34): 13910-13918.). So, the existence of the intermediate state is indeed less likely. The actual scenario is more likely that CO_3^{2-} approaches Al^{3+} , and both mutually promote hydrolysis.

Reviewer 5 provided a new potential explanation for the role of CO_3^{2-} -- a pH buffer in proximity to the Al. We believe that attributing the role of CO_3^{2-} to a dual function as a pH buffer and a catalyst can more accurately describe this process. Emphasizing either the pH buffer or catalyst alone may not be precise enough. The main reason is based on the following calculation. The K_{sp} of $\text{Al}(\text{OH})_3$ is 1.9×10^{-33} , and if we approximate the pH of the 10m LiTFSI solution to be 7 (Figure S2: 10m pH 6.82, 10m + Li_2CO_3 pH 6.87), then $\text{Al}(\text{OH})_3$ can stably exist when the concentration of Al^{3+} reaches 1.9×10^{-12} mol/L, which is quite low. It implies that even in a 10m LiTFSI solution without adding HTA, the $\text{Al}(\text{OH})_3$ passivation layer should spontaneously form after Al corrosion. However, in the XPS spectra in Figure 2, we did not observe the presence of an $\text{Al}(\text{OH})_3$ layer on the surface of Al foil after corrosion in the 10m LiTFSI despite the

pH being sufficiently high. We believe the reason is that Al^{3+} prefers to bind with TFSI rather than OH. Although both have the same charge, the abundance of TFSI is much higher than OH, making the formation of $Al(OH)_3$ challenging. After adding HTA like Li_2CO_3 , due to the higher charge and smaller size of CO_3^{2-} , Al^{3+} preferentially binds to CO_3^{2-} over TFSI. This results in that even with a higher concentration of TFSI, CO_3^{2-} can still promptly promote the hydrolysis of Al^{3+} . From this perspective, CO_3^{2-} seems more like a catalyst for catalyzing hydrolysis. At the same time, the high-potential positive electrode side will attract a large amount of CO_3^{2-} accumulation, inevitably raising the local micro-region's pH. This also provides favorable conditions for forming $Al(OH)_3$. Therefore, it is appropriate to consider it as a pH buffer for the micro-region.

Main text

"----- The other is the hydrolyzation type anodic additives (HTA), including Li_2CO_3 , Li_2SiO_3 , and $LiAlO_2$, whose anions catalyze the hydrolysis of Al^{3+} and also acts as a pH buffer in proximity to the Al foil. A more detailed discussion about the role of HTA is in Supplementary Information Note 2. -----"

Supplementary Information

"----- HTA is a catalyst for the hydrolysis of Al^{3+} and a pH buffer near Al. The reason is rooted in experimental results: $Al(OH)_3$ is not observed on Al foil after corrosion in 10m LiTFSI but observed after adding Li_2CO_3 (Figure 2E). This is not due to a significant pH difference of electrolytes in the bulk but rather arises from the catalytic action of HTA and its role in microscale pH buffering.

The K_{sp} of $Al(OH)_3$ is 1.9×10^{-33} , and if we approximate the pH of the 10 m LiTFSI solution to be 7 (Figure S2: 10 m pH 6.82, 10 m + Li_2CO_3 pH 6.87), then $Al(OH)_3$ can stably exist when the concentration of Al^{3+} reaches 1.9×10^{-12} mol/L, which is relatively low. It implies that even in a 10m LiTFSI solution without the HTA, the $Al(OH)_3$ passivation layer should spontaneously form after Al corrosion. However, in the XPS spectra in Figure 2E, the $Al(OH)_3$ layer is not observed on the surface of Al foil in the

10m LiTFSI despite the pH being sufficiently high because the Al^{3+} prefers to bind with TFSI⁻ rather than OH^- . Although both have the same charge, the abundance of TFSI⁻ is much higher than OH^- , making the formation of $\text{Al}(\text{OH})_3$ challenging.

After adding HTA like Li_2CO_3 , due to the higher charge and smaller size of CO_3^{2-} , Al^{3+} preferentially binds to CO_3^{2-} over TFSI⁻. This results in that even with a higher concentration of TFSI, CO_3^{2-} can still promptly promote the hydrolysis of Al^{3+} . From this perspective, CO_3^{2-} seems more like a catalyst for catalyzing hydrolysis. At the same time, the high-potential positive electrode side will attract a large amount of CO_3^{2-} accumulation, inevitably raising the local micro-region's pH, and providing favorable conditions for forming $\text{Al}(\text{OH})_3$. Therefore, it is appropriate to consider it as a pH buffer for the micro-region.

Q8: page10, line215. The term “pre-lithiate” usually means to lithiate active materials before the assembly of cells. However, in this study the “prelithiation” is happening at the same time of battery initial charging. Is the term “pre-lithate” appropriate in this context?

R8: Thank you very much for your comment. In the SP-ALIB pouch cell shown in Figures 6 and S15, the specific procedure involves initially charging the negative electrode of the cell at a constant voltage using an additional sacrificial Al electrode during the formation stage. After pre-lithiation, the cell is then connected back to the normal positive and negative electrodes for cycling. Therefore, we refer to this process as pre-lithiation. Your confusion likely stems from the small cell in Figure 5, where the main goal was to demonstrate that Al can provide additional capacity. The cell in Figure 5 is not the SP-ALIB structure shown in Figure 6. In this case, lithiation of TiO_2 due to Al current collector corrosion occurs simultaneously in the first cycle. However, since the platform of Al corrosion is lower than that of the positive active material LiMn_2O_4 , the additional capacity from Al corrosion occurs prior to the regular cycling.

It appears to align with the conventional definition of pre-lithiation.

Reply to the comments from Reviewer #5:

Comments: Two types of salt additive (HTA and PTA) were studied for their capabilities of forming stable passivation layer on aluminum current collector. It is identified those additives partially replace the existing native oxide layer with $\text{Al}(\text{OH})_3$ or Al-anion precipitate such as AlPO_4 and both types of layer effectively suppress pitting corrosion reaction of aluminum current collector. The extent of corrosion reaction was comparable to that of titanium current collector which is known to be highly stable under aqueous medium, which implies the possibility of the practical application of aluminum current collector for aqueous LIB.

This research doesn't seem to have significant originality in terms of research idea or strategy considering the fact that aluminum corrosion issue (in relation with passivation layer) has been well-established in non-aqueous LIB system and a lot of similar passivation-research have been conducted. However, considering the meaning of enabling the application of aluminum current collector in aqueous media and the simplicity of the method (adding $\sim 0.05\text{m}$ additive), the HTA and PTA additive strategy can be evaluated as having scientific importance and practical value worth sharing. Hence, I would support the publication of this work on the condition that following comments and questions are properly handled. Detailed comments are given below.

Reply: Thank you for your positive comments and valuable suggestions that is very helpful to improving the quality of our work.

Q1: The authors should reexamine the reaction mechanism of HTA and PTA.

Q1-1: It is claimed that TFSI anion reacts with Aluminum oxide layer and then exposed Al surface reacts with HTA or PTA anion forming passivation layer. However, considering the double-layer structure of passivation layer it also seems possible that HTA or PTA reacts with Aluminum oxide layer preferentially and form passivation layer on it, suppressing the reaction of TFSI anion.

R1-1: Thank you very much for your comment. We think that the CO_3^{2-} or PO_4^{3-} is unlikely to react with stable Al_2O_3 , and HTA or PTA is nearly impossible to involve in the passivation during the process of TFSI⁻ damaging the Al_2O_3 layer. Instead, We believe that the passivation mechanism originates from the oxidation of Al metal followed by its combination with additives rather than the destruction of Al_2O_3 to form the passivation layer.

1. As the 19th reference, Wang and coworkers proposed the mechanism for how TFSI⁻ attacks Al_2O_3 : 1. $\text{TFSI}_{\text{bulk}}^- \rightarrow \text{TFSI}_{\text{electrode}}^-$ (TFSI⁻ diffuses from bulk electrolyte to Al electrode); 2. $\text{TFSI}_{\text{electrode}}^- \rightarrow \text{TFSI}_{\text{electrode(Al)}}^-$ (TFSI⁻ absorbs onto the active spot of Al surface film); 3. $\text{TFSI}_{\text{electrode(Al)}}^- + \text{Al}_2\text{O}_3 \rightarrow [\text{Al}(\text{TFSI})_x^{3-x}]_{\text{electrode}} + \text{O}_2 + e^-$ (TFSI⁻ reacts with Al_2O_3); 4. $[\text{Al}(\text{TFSI})_x^{3-x}]_{\text{electrode}} \rightarrow [\text{Al}(\text{TFSI})_x^{3-x}]_{\text{bulk}}$ (the complex ion diffuses from Al electrode to bulk electrolyte). It implies that TFSI⁻ absorb onto the active spot and then reacts with Al_2O_3 . The reaction product is $\text{Al}(\text{TFSI})_x^{3-x}$ instead of Al^{3+} . It is much harder for CO_3^{2-} or PO_4^{3-} to combine with $\text{Al}(\text{TFSI})_x^{3-x}$ because $\text{Al}(\text{TFSI})_x^{3-x}$ gets much larger, and the quantity of positive charges is less than Al^{3+} . Our core mechanism is that when Al metal is oxidized to Al^{3+} , CO_3^{2-} or PO_4^{3-} has a smaller volume and more negative charges than TFSI⁻, thus preferring to bind with Al^{3+} over TFSI⁻. However, when TFSI⁻ attacks Al_2O_3 , TFSI⁻ is undoubtedly the primary participant in forming reaction products instead of CO_3^{2-} or PO_4^{3-} . Thus, we believe the passivation mechanism originates from the oxidation of Al metal instead of Al_2O_3 .

2. Through calculations, we demonstrated that the initial amount of Al_2O_3 is insufficient to provide the pre-lithiation capacity shown in Figure R2 (Figure S12). Therefore, the corrosion-passivation process should mainly be contributed by Al metal. As we discussed above, the process of TFSI⁻ attacking Al_2O_3 is also an electrochemical oxidation reaction-- $\text{TFSI}_{\text{electrode(Al)}}^- + \text{Al}_2\text{O}_3 \rightarrow [\text{Al}(\text{TFSI})_x^{3-x}]_{\text{electrode}} + \text{O}_2 + e^-$. In this process, oxygen in Al_2O_3 undergoes oxidation reactions, and from this, we can calculate the corresponding Al_2O_3 's theoretical specific

capacity: $\frac{3 \times 2 \times 96485}{3.6 \times 10^2} \text{ mAh/g} = 1576 \text{ mAh/g}$. The area of the Al foil is 2 cm^2 for the battery in Figure R2 (Figure S12), and we assume the surface area is 4 cm^2 . The original Al_2O_3 is less than 10 nm based on the XPS results in Figure 2, which show that the pristine Al sample showed Al metal's peaks after 60 s Ar^+ etching. The density of Al_2O_3 is 3.5 g/cm^3 . Therefore, the max capacity that Al_2O_3 could contribute is $3.5 \times 4 \times 10 \times 10^{-7} \times 1576 = 0.02 \text{ mAh}$. As shown in Figure R2 (Figure S12), the corrosion of Al contributes about 0.07 mAh capacity ($55 \text{ mAh/g} \times 1.284 \text{ mg}$, 55 mAh/g is read from Figure S12 and 1.284 mg is the mass of TiO_2 to calculate the specific capacity of the cell). This implies that even if all the Al_2O_3 attached to both sides of the Al current collector is oxidized, it can only provide a capacity of 0.02 mAh , much less than the actual capacity of 0.07 mAh from Al in the first cycle. Therefore, the primary reactant in the corrosion-passivation process should be Al metal rather than surface Al_2O_3 . Moreover, from the XPS results with the addition of HTA or PTA in Figure 2, after the surface $\text{Al}(\text{OH})_3$ or AlPO_4 layer is etched for 60 s , there is a distinct layer of Al_2O_3 underneath, indicating that Al foil has completed passivation before TFSI destroys the Al_2O_3 layer. We are more inclined to believe that this is due to TFSI attacking active sites, and after internal Al metal corrosion, Al^{3+} diffuses and combines with HTA or PTA to form a passivation layer around the pitting sites. During the passivation process, the Al_2O_3 layer of the undamaged region around the pitting site is covered.

Main text

"----- It implies that TFSI is impossible to ceaselessly destroy the Al_2O_3 layer due to the passivation of Al foil because once the corrosion pitting sites occur, the dissolving Al^{3+} immediately reacts with HTA or PTA, resulting in the undamaged Al_2O_3 layer is covered around the pitting site. -----"

Q1-2: On a related note, if the reaction between TFSI anion and Aluminum oxide layer

has to occur first for the reaction between Al(s) and additive anion to occur, could the effects of HTA and PTA be insignificant or absent in salt systems other than LiTFSI?

RI-2: Thank you very much for your comment.

Firstly, we would like to clarify our understanding of the electrochemical corrosion of Al in the electrolyte. We believe it implies that Al is oxidized at active sites, and its corrosion products diffuse into the solution, leaving the reaction site and creating a localized loss.

The salts can be categorized into two groups: one that cannot destroy the Al₂O₃ layer in its aqueous solution and another that cannot. For the former, if a salt cannot even corrode the Al₂O₃ layer, the internal Al metal cannot be oxidized, penetrate the dense oxide layer, and diffuse into the solution. In other words, the aqueous solution of this salt is incapable of corroding Al because its original Al₂O₃ is naturally a good protection layer. In such a scenario, we can directly use this electrolyte without additional anti-corrosion additives like HTA or PTA. For the latter, if a salt could destroy the Al₂O₃ layer, then there is a chance for the reaction between Al(s) and the additive anion to occur. In this case, we should consider two things. First, the priority is the anions of HTA/PTA and the salt in binding with Al³⁺. Second, the salt can destroy Al(OH)₃ or AlPO₄, continuing to corrode the Al current collector. From the literature that the Reviewer 2 mentioned, PO₄³⁻ increased the pitting corrosion potential in the Cl⁻ solutions (Rudd W J, Scully J C. The function of the repassivation process in the inhibition of pitting corrosion on aluminium[J]. Corrosion Science, 1980, 20(5): 611-631.) while HPO₄²⁻ decreased the dissolution rate of Al₂O₃ in HCl solutions (Abd El Aal E E, Abd El Wanees S, Farouk A, et al. Factors affecting the corrosion behaviour of aluminium in acid solutions. II. Inorganic additives as corrosion inhibitors for Al in HCl solutions[J]. Corrosion Science, 2013, 68: 14-24.). Therefore, the mechanism of PTA or HTA should be applicable in other salt systems as well.

Q1-3 : It is claimed that HTA anion forms aluminum-anion intermediate before hydrolyzation. It is acceptable the presence of HTA is necessary for the hydrolyzation to occur according to the XPS analysis result. However, it is not clear if HTA forms aluminum-anion intermediate. Or is it not the role of the HTA anion to serve as a pH buffer in proximity to the aluminum

RI-3: Thank you very much for your good advice. In this version, we more clearly interpreted the mechanism involving HTA and updated the description of the role of the HTA anion.

Initially, we used $Al_2(CO_3)_3$ as an intermediate state to help readers better understand the reaction scenario we wanted to portray, similar to PTA. The core of the passivation reaction is that the anions of the additive bind preferentially to Al^{3+} over TFSI anions. However, the existence of Al_2O_3 has not been accurately confirmed until 2023 when Lkhamsuren Bayarjargal synthesized $Al_2(CO_3)_3$ under high CO_2 pressure of 24-28 GPa (Anhydrous Aluminum Carbonates and Isostructural Compounds[J]. Inorganic Chemistry, 2023, 62(34): 13910-13918.). So, the existence of the intermediate state is indeed less likely. The actual scenario is more likely that CO_3^{2-} approaches Al^{3+} , and both mutually promote hydrolysis. The representation of the corresponding equation in Figure 1 should be: $Al^{3+}(aq) + CO_3^{2-}(aq) + H_2O \rightarrow Al(OH)_3 + H_2CO_3$, instead of $Al^{3+}(aq) + CO_3^{2-}(aq) \rightarrow Al_2(CO_3)_3$ and $Al_2(CO_3)_3 + H_2O \rightarrow Al(OH)_3 + H_2CO_3$.

Your comments on its role as a pH buffer are helpful in aiding reader understanding. After careful consideration and calculations, we believe that emphasizing both the pH buffer and catalyst roles of HTA provide a more accurate description of the mechanism. Emphasizing either the pH buffer or catalyst alone may not be precise enough. The main reason is based on the following calculation. The K_{sp} of $Al(OH)_3$ is 1.9×10^{-33} and if we approximate the pH of the 10m LiTFSI solution to be 7 (Figure S2: 10m pH 6.82, 10m + Li_2CO_3 pH 6.87), then $Al(OH)_3$ can stably exist when the concentration of

Al^{3+} reaches 1.9×10^{-12} mol/L, which is quite low. This implies that even in a 10m LiTFSI solution without the addition of HTA, a spontaneous formation of $Al(OH)_3$ passivation layer should occur after Al corrosion. However, in the XPS spectra in Figure 2, we did not observe the presence of an $Al(OH)_3$ layer on the surface of Al foil after corrosion in the 10m LiTFSI despite the pH being sufficiently high. We believe the reason is that Al^{3+} prefers to bind with TFSI rather than OH. Although both have the same charge, the abundance of TFSI is much higher than OH, making the formation of $Al(OH)_3$ challenging. After adding HTA like Li_2CO_3 , due to the higher charge and smaller size of CO_3^{2-} , Al^{3+} preferentially binds to CO_3^{2-} over TFSI. This results in that even with a higher concentration of TFSI, CO_3^{2-} can still promptly promote the hydrolysis of Al^{3+} . From this perspective, CO_3^{2-} seems more like a catalyst for catalyzing hydrolysis. At the same time, the high-potential positive electrode side will attract a large amount of CO_3^{2-} accumulation, inevitably raising the local micro-region's pH. This also provides favorable conditions for the formation of $Al(OH)_3$. Therefore, it is appropriate to consider it as a pH buffer for the micro-region.

Figure 1 has been replaced with the equation changed to $Al^{3+}(aq) + CO_3^{2-}(aq) + H_2O \rightarrow Al(OH)_3 + H_2CO_3$.

Main text

"----- The other is the hydrolyzation type anodic additives (HTA), including Li_2CO_3 , Li_2SiO_3 , and $LiAlO_2$, whose anions catalyze the hydrolysis of Al^{3+} and also acts as a pH buffer in proximity to the Al foil. A more detailed discussion about the role of HTA is in Supplementary Information Note 2.-----"

Supplementary Information

"----- HTA is a catalyst for the hydrolysis of Al^{3+} and a pH buffer near Al. The reason is rooted in experimental results: $Al(OH)_3$ is not observed on Al foil after corrosion in 10m LiTFSI but observed after adding Li_2CO_3 (Figure 2E). This is not due to a

significant pH difference of electrolytes in the bulk but rather arises from the catalytic action of HTA and its role in microscale pH buffering.

The K_{sp} of $\text{Al}(\text{OH})_3$ is 1.9×10^{-33} , and if we approximate the pH of the 10 m LiTFSI solution to be 7 (Figure S2: 10 m pH 6.82, 10 m + Li_2CO_3 pH 6.87), then $\text{Al}(\text{OH})_3$ can stably exist when the concentration of Al^{3+} reaches 1.9×10^{-12} mol/L, which is relatively low. It implies that even in a 10m LiTFSI solution without the HTA, the $\text{Al}(\text{OH})_3$ passivation layer should spontaneously form after Al corrosion. However, in the XPS spectra in Figure 2E, the $\text{Al}(\text{OH})_3$ layer is not observed on the surface of Al foil in the 10m LiTFSI despite the pH being sufficiently high because the Al^{3+} prefers to bind with TFSI^- rather than OH^- . Although both have the same charge, the abundance of TFSI^- is much higher than OH^- , making the formation of $\text{Al}(\text{OH})_3$ challenging.

After adding HTA like Li_2CO_3 , due to the higher charge and smaller size of CO_3^{2-} , Al^{3+} preferentially binds to CO_3^{2-} over TFSI^- . This results in that even with a higher concentration of TFSI^- , CO_3^{2-} can still promptly promote the hydrolysis of Al^{3+} . From this perspective, CO_3^{2-} seems more like a catalyst for catalyzing hydrolysis. At the same time, the high-potential positive electrode side will attract a large amount of CO_3^{2-} accumulation, inevitably raising the local micro-region's pH, and providing favorable conditions for forming $\text{Al}(\text{OH})_3$. Therefore, it is appropriate to consider it as a pH buffer for the micro-region.

-----"

Q2: The over-lithiation of LMO occurs at low voltage level. Comparing the two graphs in Figure 3A, the over-lithiation strategy also has a drawback since it results in the cell's operating voltage being lowered while the capacity remains the same. In a practical battery system where energy should be provided at a constant voltage level, such design could be a serious disadvantage rather than an advantage. The authors need to present

a balanced and comprehensive perspective that takes this into account. In addition, it is recommended to present comparative cycling stability data of “Al current collector, HTA or PTA additive” vs “Ti current collector” under constant high voltage system (no over-lithiation).

R2: Thank you very much for your advice. Figure S18 (Figure R5) will be added in the Supplementary Information to show that the capacity of the over-lithiated LiMn_2O_4 is minimal and has little impact on energy density. Besides, Figure S19 (Figure R6) shows the cycling stability of Al current collector with HTA additive vs Ti current collector (no over-lithiation) and we will also add it into Supplementary Information.

Figure 3 is intended to illustrate two key points: firstly, Al undergoes oxidation during the battery charging process, and secondly, the corrosion of Al can lead to the over-lithiation of LiMn_2O_4 , providing a potential avenue to turn Al corrosion into a valuable resource (electron supplement). To present this phenomenon vividly to readers, we utilized a minimal amount of active material. Additionally, we placed an additional 2 cm^2 Al foil directly behind the single-layer LiMn_2O_4 electrode to amplify the effect of Al oxidation during charging, allowing it to be visible in the curves. We have discussed this question detailly in our response to the question 3 from the second reviewer. As shown in Figure R4B, the blue line is from the normal $\text{LiMn}_2\text{O}_4\text{-TiO}_2$ battery without sacrificial Al electrode and anti-corrosion additive, just the same condition as the battery in Figure 5A, but the electrochemical curve is quite different without the lower over-lithiation plateau. That is not because Al is not corroded at a big battery but because the corrosion contributes too little capacity compared to active material to show up in the curve. Therefore, in large batteries, the corrosion of Al current collectors does not provide significant electron supplementary. It cannot even surpass the capacity loss caused by the formation of the SEI and hydrogen evolution at the negative electrode during the first cycle, making it unable to induce LiMn_2O_4 over-lithiation.

As for SP-ALIB, benefit from the corrosion of the sacrificial Al current collector in the

formation process, LiMn_2O_4 get a little over-lithiated in the first cycle, which means that after offsetting the capacity loss caused by the formation of the SEI and hydrogen evolution during the first cycle, we can still retain a portion of additional lithium sources to support longer cycles. We do not compensate excessively, so it hardly affects energy density and the over-lithiation platform introduced by Al oxidation will gradually be consumed by side reactions during cycling. It can be observed that from the first to the fifth cycle, the over-lithiation platform is gradually decreasing in Figure R5.

Cycling stability comparison of Al current collector with Li_2CO_3 and Ti current collector is shown in Figure R6. There is no over-lithiation shown in the first cycle. We speculate that the capacity compensation from the electrochemical passivation of the Al current collector is not sufficient to offset the capacity loss caused by SEI formation and hydrogen evolution side reactions because there is no additional Al foil and the third sacrificial Al electrode with specific formation process. Therefore, it may not be reflected in the curves (over-lithiation), but it seems to play some role.

Figure R4. Different effects of Al current collector corrosion on low-capacity and high-capacity batteries. (A) small pouch cells with a capacity of about 0.2mAh. From Figure S12. (B) 0.5Ah stacked pouch cell from Figure 6. (C) the schematic diagram of the self-prolonging ALIBs.

Figure R5. The actual over-lithiation plateau in 0.5 Ah $\text{LiMn}_2\text{O}_4\text{-TiO}_2$ pouch cell. This SP-ALIB battery is pre-lithiated by the sacrificial pre-lithiation Al electrode first, and then cycled. 10m LiTFSI + Li_2CO_3 .

Figure R6. Cycling stability of Al current collector with 10m LiTFSI + Li₂CO₃ vs Ti current collector with 10m LiTFSI. (A) Cycling stability of LiMn₂O₄-TiO₂ batteries. (B) The charge-discharge profile of the first cycle for the battery with Al current collector using 10m + Li₂CO₃. No additional Al foil or the sacrificial Al electrode.

We explained the impact of energy density from the over-lithiation plateau in Supplementary Information Note 4 and compared the cycling stability of Al + Li₂CO₃ with the Ti current collector in Figure S19.

Supplementary Information

"-----" We do not excessively get TiO₂ pre-lithiated in the formation process, so after offsetting the side reactions of SEI formation and hydrogen evolution, the excess capacity of over-lithiated LiMn₂O₄ is insignificant, as shown in Figure S18. Though the voltage plateau of the over-lithiated LiMn₂O₄ is low, it does not impact the battery's energy density much. -----"

"-----" Cycling stability comparison of Al current collector with Li₂CO₃ and Ti current collector is shown in Figure S19. There is no over-lithiation shown in the first cycle. The capacity compensation from the electrochemical passivation of the Al current collector is not sufficient to offset the capacity loss caused by SEI formation and hydrogen evolution side reactions because there is no additional Al foil or the third sacrificial Al electrode with specific formation process. Therefore, it may not be

reflected in the curves (over-lithiation), but it seems to play some role. 
REVIEWER COMMENTS

Reviewer #5 (Remarks to the Author):

Attached separately.

This work is primarily concerned with the corrosion inhibition reaction proceeding in a sequential manner. The precise confirmation of the corrosion inhibition mechanism is further required since it is also related to the over-lithiation of lithium manganese oxide (LMO) and the pre-lithiation of the Titanium dioxide anode. The explanation of this mechanism involves considerable complexity since individual or interactive actions of various ionic species (TFSI⁻, PO₄³⁻, HPO₄²⁻, H₂PO₄⁻, CO₃²⁻, HCO₃⁻, hydronium/hydroxide ion, Al³⁺) must be taken into account. While the authors have proposed an interpretation based on the interaction of TFSI anion with the Al₂O₃ layer and the XPS data presented in this work, it seems uncertainties still remain.

1. Specifically, regarding the corrosion inhibition mechanism of HTA, the authors claim that carbonate anions (CO₃²⁻) can preferentially coordinate with Al³⁺ (in comparison with TFSI⁻ anion) and serves a catalytic role based on the fact that they are small and divalent. However, pH around 7 implies the presence of carbonate salt species in the monovalent form (HCO₃⁻) among the potential candidates; CO₃²⁻, HCO₃⁻, H₂CO₃. While the smaller size of bicarbonate anion (HCO₃⁻) would mean smaller steric hindrance, and thus easier access toward Al³⁺, it also should be considered that TFSI outnumbers HCO₃⁻ in a ratio of 100:1. The fact that TFSI⁻ anion reacts with Al₂O₃, decomposing it and forming ionic species such as [Al(N(SO₂CF₃)₂)_x]^{3-x}, also seems to indicate the high stability of the complex. Given these circumstances, additional validation is deemed necessary for the assertion of carbonate species' preferential coordination and its role as catalyst.

2. On a related note, the role of the bicarbonate ions as pH buffer, more specifically, their capability to combine with proton, appears to be a more crucial factor.

2-1. The formation of Aluminum hydroxide would result in an abrupt pH decrease in close proximity to the surface of Al₂O₃, which in turn would suppress further formation of the hydroxide. Several previous research has corroborated the abrupt and localized pH changes around the electrode interface, in case where proton and hydroxide ion participate in chemical and/or electrochemical reactions. The bicarbonate ion might enable the formation of Al(OH)₃ without localized pH changes (**reaction (1)**) or combines with H⁺ near the electrode interface to mitigate pH fluctuations, facilitating additional Al(OH)₃ formation (**reaction (2)**).

2-2. The migration or diffusion phenomenon during charge process also support the buffer role hypothesis. The electrical bias during the charging process induces the approach of the bicarbonate ions towards the Al electrode. In contrast, neutral H₂CO₃ would diffuse into bulk electrolyte. The HCO₃⁻ ions can be replenished near the electrode, maintaining its role. The diffused H₂CO₃ will dissociate back into HCO₃⁻ in the bulk electrolyte. The pH variation during this dissociation process is expected to be gradual. While the recovery of bicarbonate ion supports the catalytic role of carbonate salt species, it might be irrelevant with coordination of Al³⁺.

2-3. HCO₃⁻ and HPO₄²⁻ ions are capable of binding with H⁺, whereas the TFSI⁻ anion (the conjugate base of strong acid HTFSI) has a significantly lower tendency to do so. The huge difference in the tendency of the conjugate anions to engage in interaction with protons

might explain the reason HCO_3^- and HPO_4^{2-} ions could be exceptionally functional despite being ~ 100 times less in quantity, further supporting the buffer role interpretation. The electrolyte pH data showing that a minor amount of additive can significantly increase the electrolyte pH also supports the effectivity of additive anion in binding with H^+ and raising OH^- concentration.

3. A major concern lies in the fact that HPO_4^{2-} can also bind with H^+ and it seems it can function similarly with the HTA. Closer inspection of XPS data reveals that the peaks of $\text{Al}(\text{OH})_3$ and $\text{Al}(\text{PO}_4)_3$ are not distinctly separated. XPS P 2p spectra or other analyses demonstrating the presence of $\text{Al}(\text{PO}_4)_3$ are required.

Overall, the authors addressed most of comments or questions properly. However, as mentioned, several uncertainties still seem to remain especially regarding the corrosion inhibition mechanism. The explanation supporting the preferential coordination of carbonate ion species with Al^{3+} seems to rather inaccurate or lack solid evidences whereas buffer role hypothesis seems to better explain the phenomenon and thus requires more attention. Additionally, it seems extremely important to reexamine the composition of passivation layer formed by PTA additives. While this work has scientific importance and practical impact worth sharing, thorough examination on reaction mechanism and appropriate interpretation have to be supplemented.

Reply to the comments from Reviewer #5:

Comments: This work is primarily concerned with the corrosion inhibition reaction proceeding in a sequential manner. The precise confirmation of the corrosion inhibition mechanism is further required since it is also related to the over-lithiation of lithium manganese oxide (LMO) and the pre-lithiation of the Titanium dioxide anode. The explanation of this mechanism involves considerable complexity since individual or interactive actions of various ionic species (TFSI⁻, PO₄³⁻, HPO₄²⁻, H₂PO₄⁻, CO₃²⁻, HCO₃⁻, hydronium/hydroxide ion, Al³⁺) must be taken into account. While the authors have proposed an interpretation based on the interaction of TFSI anion with the Al₂O₃ layer and the XPS data presented in this work, it seems uncertainties still remain.

Overall, the authors addressed most of comments or questions properly. However, as mentioned, several uncertainties still seem to remain especially regarding the corrosion inhibition mechanism. The explanation supporting the preferential coordination of carbonate ion species with Al³⁺ seems to rather inaccurate or lack solid evidences whereas buffer role hypothesis seems to better explain the phenomenon and thus requires more attention. Additionally, it seems extremely important to reexamine the composition of passivation layer formed by PTA additives. While this work has scientific importance and practical impact worth sharing, thorough examination on reaction mechanism and appropriate interpretation have to be supplemented.

Reply: Thank you for your valuable suggestions on our manuscript. Your insights have greatly inspired our re-evaluation of the mechanism and lead us to identify certain loopholes in our initial thought process. We sincerely express our gratitude to the reviewers in the acknowledgment section for your valuable discussions.

Q1: Specifically, regarding the corrosion inhibition mechanism of HTA, the authors claim that carbonate anions (CO₃²⁻) can preferentially coordinate with Al³⁺ (in comparison with TFSI⁻ anion) and serves a catalytic role based on the fact that they

are small and divalent. However, pH around 7 implies the presence of carbonate salt species in the monovalent form (HCO_3^-) among the potential candidates; CO_3^{2-} , HCO_3^- , H_2CO_3 . While the smaller size of bicarbonate anion (HCO_3^-) would mean smaller steric hindrance, and thus easier access toward Al^{3+} , it also should be considered that TFSI outnumbered HCO_3^- in a ratio of 100:1. The fact that TFSI⁻ anion reacts with Al_2O_3 , decomposing it and forming ionic species such as $[\text{Al}(\text{N}(\text{SO}_2\text{CF}_3)_2)_x]^{3-x}$, also seems to indicate the high stability of the complex. Given these circumstances, additional validation is deemed necessary for the assertion of carbonate species' preferential coordination and its role as catalyst.

R1: Thank you very much for pointing out the possible effect of the monovalent form (HCO_3^-). The mechanism of HTA in the previous vision is based on the interaction of CO_3^{2-} and Al^{3+} . However, we overlooked the basic fact that HCO_3^- dominates at a pH around 7, instead of CO_3^{2-} . We discussed the possible mechanism in the manuscript and updated Figure 1 accordingly and deleted the discussion about the preference of HTA anions to bind with Al^{3+} .

Regarding the question of why HCO_3^- preferentially binds with Al^{3+} , we conducted some calculations. However, the results yielded some surprisingly conclusions, implying the complexity of the mechanism of HTA.

Firstly, as shown in Figure R1, HCO_3^- dominates rather than CO_3^{2-} in aqueous solution with pH around 7. We aim to calculate the precise concentration of HCO_3^- in the solution after adding Li_2CO_3 until saturation in 10m LiTFSI.

Assuming $[\text{HCO}_3^-] = a$, $[\text{CO}_3^{2-}] = b$, $[\text{H}_2\text{CO}_3] + [\text{HCO}_3^-] + [\text{CO}_3^{2-}] = y$ and $[\text{H}^+] = c$. At the molality of 10m of LiTFSI, it is approximately equivalent to the molarity of 4M LiTFSI. Thus, $[\text{Li}^+] = 4 + 2y$ because all the H_2CO_3 , HCO_3^- and CO_3^{2-} come from Li_2CO_3 and we assume that the volume of the electrolyte doesn't change much due to the dissolution of Li_2CO_3 .

$$(4 + 2y)^2 \times b = 2.5 \times 10^{-2} \quad (4)$$

$$\frac{ac}{y-a-b} = 4.3 \times 10^{-7} \quad (5)$$

$$\frac{bc}{a} = 4.8 \times 10^{-11} \quad (6)$$

We can get (4) from (1), get (5) from (2) and get (6) from (3). Given that $[\text{H}^+]$ (c) can be determined through pH measurements, the three equations can be solved to obtain the values of 'a', 'b', and 'y'. From Equation (6) we get $a = \frac{bc}{4.8 \times 10^{-11}}$. Substituting 'a' into Equation (5), we obtain $b = \frac{y}{1 + \frac{c^2}{20.64 \times 10^{-18}} + \frac{c}{4.8 \times 10^{-11}}}$. Substituting 'b' into Equation (4), we obtain that $(4 + 2y)^2 y = (1 + \frac{c^2}{20.64 \times 10^{-18}} + \frac{c}{4.8 \times 10^{-11}}) \times 2.5 \times 10^{-2}$. Assume pH = x, then $c = 10^{-x}$. Finally, we draw the Figure R2 based on the result. This curve illustrates the relationship of ' $[\text{H}_2\text{CO}_3] + [\text{HCO}_3^-] + [\text{CO}_3^{2-}]$ ' and pH of 10m LiTFSI with saturated Li_2CO_3 .

Surprisingly, when pH is around 7, $[\text{H}_2\text{CO}_3] + [\text{HCO}_3^-] + [\text{CO}_3^{2-}]$ is larger than 1 mol/L. It's quite impossible, because $[\text{Li}^+] = 4+2y$, it exceeds 6M, while LiTFSI saturates at around 5M. From another aspect, Li_2CO_3 is slightly soluble in water and 10m LiTFSI should not be able to dissolve so much Li_2CO_3 . In fact, from our experiment, the saturation concentration of Li_2CO_3 is less than 0.05m. Here is a paradox.

As shown in Figure R2, if the pH of 10m LiTFSI with saturated Li_2CO_3 is higher, for example reaching 8 or 9, the calculation and experiment are in agreement. Therefore,

we examined the pH of the solution again, but it aligns closely with the data we previously measured. The pH of 10m LiTFSI with saturated Li_2CO_3 is still around 7. The calculation implies that the 10m LiTFSI is able to dissolve a lot of Li_2CO_3 or the 10m LiTFSI with saturated Li_2CO_3 has a higher pH than 8. However, the experimental fact is that 10m LiTFSI dissolves a little Li_2CO_3 and its pH only increases a little at around 7.

The disparity between the aforementioned calculations and experimental results indicates the complexity of the underlying mechanism. We can only deduce possible mechanisms based on the experimental outcomes. It is known that adding Li_2CO_3 to 10m LiTFSI results in the formation of $\text{Al}(\text{OH})_3$ on the surface after the oxidation of Al at high potential. Additionally, at a pH of approximately 7, HCO_3^- predominantly exist in the solution among carbonate species. Therefore, the most plausible mechanism, as you mentioned in Q2-1, is $\text{Al}^{3+} + 3\text{HCO}_3^- + 3\text{H}_2\text{O} = \text{Al}(\text{OH})_3 + 3\text{H}_2\text{CO}_3$. To ensure a more rigorous argument, we have removed the discussion about the preference of HTA to bind with Al^{3+} , as it lacked solid theoretical support.

Figure R1. Distribution of carbonaceous species in water solution with pH at different total dissolved carbon concentrations (TC) (TC: solid line, 0.5 mol L^{-1} ; dashed line, 0.1 mol L^{-1} ; temperature, $25 \text{ }^\circ\text{C}$) (Reprinted with permission from Zhong H, Fujii K, Nakano Y, et al. Effect of CO_2 bubbling into aqueous solutions used for electrochemical reduction of CO_2 for energy conversion and storage[J].

Figure R2. The variation of $[H_2CO_3] + [HCO_3^-] + [CO_3^{2-}]$ with pH in 10m LiTFSI with saturated Li_2CO_3 . The curve is based on calculation, instead of experiment.

Main text "-----"

(reaction path: (iii) $Al(s) - e^- + H_xM^{n+}(aq) + H_2O \rightarrow Al(OH)_3(s) + H_{x+1}M^{1-n}(aq)$).

-----"

Supplementary Information "-----"

The pH data from Figure S2 indicate a slight increase upon the addition of HTA. This is due to the ability of HTA to bind with H^+ (as pH buffer). Therefore, when the working electrode is at a high potential, the anions of HTA aggregate near the electrode, potentially amplifying this ability and facilitating the formation of $Al(OH)_3$.

-----"

Q2: On a related note, the role of the bicarbonate ions as pH buffer, more specifically,

their capability to combine with proton, appears to be a more crucial factor.

Q2-1: The formation of Aluminum hydroxide would result in an abrupt pH decrease in close proximity to the surface of Al₂O₃, which in turn would suppress further formation of the hydroxide. Several previous research has corroborated the abrupt and localized pH changes around the electrode interface, in case where proton and hydroxide ion participate in chemical and/or electrochemical reactions. The bicarbonate ion might enable the formation of Al(OH)₃ without localized pH changes (reaction (1)) or combines with H⁺ near the electrode interface to mitigate pH fluctuations, facilitating additional Al(OH)₃ formation (reaction (2)).

Q2-2: The migration or diffusion phenomenon during charge process also support the buffer role hypothesis. The electrical bias during the charging process induces the approach of the bicarbonate ions towards the Al electrode. In contrast, neutral H₂CO₃ would diffuse into bulk electrolyte. The HCO₃⁻ ions can be replenished near the electrode, maintaining its role. The diffused H₂CO₃ will dissociate back into HCO₃⁻ in the bulk electrolyte. The pH variation during this dissociation process is expected to be gradual. While the recovery of bicarbonate ion supports the catalytic role of carbonate salt species, it might be irrelevant with coordination of Al³⁺.

Q2-3: HCO₃⁻ and HPO₄²⁻ ions are capable of binding with H⁺, whereas the TFSI⁻ anion (the conjugate base of strong acid HTFSI) has a significantly lower tendency to do so. The huge difference in the tendency of the conjugate anions to engage in interaction with protons might explain the reason HCO₃⁻ and HPO₄²⁻ ions could be exceptionally functional despite being ~100 times less in quantity, further supporting the buffer role interpretation. The electrolyte pH data showing that a minor amount of additive can significantly increase the electrolyte pH also supports the effectivity of

additive anion in binding with H^+ and raising OH^- concentration.

R2: Thank you very much for your advice and helpful analysis on the possible mechanism. We designed experiments to investigate the pH changes near the electrode, aiming to validate potential mechanisms. The results imply that the mechanism could be the equation (1) you proposed in Q2-1, as this approach does not lead to drastic changes in the microscale pH. And your explanation of the mutual conversion between carbonate and bicarbonate to maintain catalytic activity in Q2-2 is highly credible. We removed the relevant discussion about the preference of HTA anions to bind with Al^{3+} , replaced the Figure 1 and discussed the possible mechanism in the main text and supplementary information.

We choose five pH indicators: phenolphthalein, phenol red, bromothymol blue, methyl red and methyl orange. The overall colorimetric pH range is from acidic to alkaline conditions (3.1-10). The colour of the 10m LiTFSI and 10m LiTFSI + Li_2CO_3 is compared by adding pH indicators in the standing state as in Figure R3. Though, the addition of Li_2CO_3 resulted in slight pH increase, it did not induce noticeable colour variations as expected. However, we may be able to see the colour change when electrochemical experiment is being conducted if there is a drastic pH change near the electrode. Moreover, this can help us validate the pH buffer function of HTA. The pH data from Figure S2 indicate a slight increase in solution with HTA. Therefore, when the working electrode is at a high potential, the accumulation of HTA anions near the electrode may amplify their pH buffering effect, possibly observable through changes in colour.

The same CA experiment is done as Figure 2. Photos were taken while the working electrode is at 4.5V vs Li/Li^+ in 10m LiTFSI + Li_2CO_3 . Figure R3A, R3B and R3C illustrate the consistent result that HER is happening at the counter electrode, thus the local pH changes drastically. The counter electrode turns red with phenolphthalein and phenol red and turns blue with bromothymol blue. These experimental results

validate the rationality and feasibility of our experimental design. If we can observe significant pH changes near the working electrode, it might guide us to discover a more accurate mechanism for Al corrosion and passivation. However, in all five sets of experiments, we did not observe the similar discoloration phenomena near the working electrode. The results imply that pH changes near the working electrode at high potential are subtle. Therefore, the more possible mechanism is that HCO_3^- enables the formation of $\text{Al}(\text{OH})_3$ without much localized pH changes, just as you mentioned in Q2-1. However, we are aware that this experimental design is somewhat rudimentary, and the colour change of the pH indicator may not be highly sensitive. Visual observation also introduces a considerable margin of error, requiring relatively significant changes to obtain more accurate results. Therefore, we can only make reasonable inferences about the mechanism based on the results.

Overall, we are inclined to believe that HCO_3^- plays its role according to $\text{Al}^{3+} + 3\text{HCO}_3^- + 3\text{H}_2\text{O} = \text{Al}(\text{OH})_3 + 3\text{H}_2\text{CO}_3$, leading to the formation of $\text{Al}(\text{OH})_3$ on the surface of Al at high potential. Although we did not directly observe pH changes near the working electrode, the pH data from Figure S2 indicate a slight increase upon the addition of HTA. This is due to the ability of HTA to bind with H^+ (as pH buffer). Therefore, when the working electrode is at a high potential, the anions of HTA aggregate near the electrode, potentially amplifying this ability. While the lack of sensitivity in the pH indicator and other factors may have prevented the direct observation of the pH buffer effect of HTA anions near the working electrode, we reasonably infer that its mechanism is indeed as described, thus facilitating the formation of $\text{Al}(\text{OH})_3$.

Main text "-----"

slightly soluble lithium weak acid salts like Li_3PO_4 , Li_2CO_3 , Li_2SiO_3 , and LiAlO_2 are proposed as a new passivation additive for Al metal in an aqueous electrolyte, whose anions after hydrolysis can combine with proton, playing a role like a pH buffer in proximity to the Al foil and facilitating the formation of $\text{Al}(\text{OH})_3$.

Moreover, in Supplementary Information Note 2, we discussed the influence of the amounts of HTA on the corrosion-resistance.

-----"

Supplementary Information "-----

The pH data from Figure S2 indicate a slight increase upon the addition of HTA. This is due to the ability of HTA to bind with H^+ (as pH buffer). Therefore, when the working electrode is at a high potential, the anions of HTA aggregate near the electrode, potentially amplifying this ability and facilitating the formation of $Al(OH)_3$.

Take HCO_3^- for example, it tends to play a catalytic role. The charging process attracts HCO_3^- to the Al current collector at positive electrode. HCO_3^- assists in the hydrolysis of Al^{3+} to form $Al(OH)_3$, while itself transforms into H_2CO_3 and diffuses into the bulk electrolyte. Corresponding equation is: $Al^{3+}(aq) + HCO_3^-(aq) + H_2O \rightarrow Al(OH)_3 + H_2CO_3$. Then the diffused H_2CO_3 dissociate back to HCO_3^- in the bulk electrolyte to maintain the catalytic activity of HCO_3^- . Therefore, HTA should not be limited by the current quantity. Therefore, there is no need to worry about the low content of HTA additive affecting the passivation effect. It is not consumed. It mainly serves as a catalyst and pH buffer.

-----"

Figure R3. The microscale pH variation near the electrode. (A) 10m LiTFSI / 10m LiTFSI + Li_2CO_3 adding phenolphthalein. (B) Add phenol red. (C) Add bromothymol blue. (D) Add methyl red. (E) Add methyl orange. The left bottle contains 10m LiTFSI,

while the red bottle contains 10m LiTFSI + Li₂CO₃. The left electrode is working electrode at 4.5V vs Li/Li⁺, the right electrode is counter electrode and the electrolyte is 10m LiTFSI + Li₂CO₃ with different pH indicators. Both electrodes are Al foil.

Q3: A major concern lies in the fact that HPO₄²⁻ can also bind with H⁺ and it seems it can function similarly with the HTA. Closer inspection of XPS data reveals that the peaks of Al(OH)₃ and AlPO₄ are not distinctly separated. XPS P 2p spectra or other analyses demonstrating the presence of AlPO₄ are required.

R3: Thank you very much for pointing out our mistake. After further verification, we have found that the passivation layer formed by Li₃PO₄ is indeed Al(OH)₃. Your judgment and analysis are extremely accurate, and it holds significant importance for our manuscript. We have removed all discussions regarding PTA, made modifications to Figure 1 and Figure 2 and categorized Li₃PO₄ into the HTA as well.

We overlooked the fact that H₂PO₄⁻ and HPO₄²⁻ dominate rather than PO₄³⁻ at pH around 7 as shown in Figure R4. The more possible reactions are: Al³⁺ + 3H₂PO₄⁻ + 3H₂O = Al(OH)₃ + 3H₃PO₄ ① and Al³⁺ + 3HPO₄²⁻ + 3H₂O = Al(OH)₃ + 3H₂PO₄⁻ ②.

In fact, the XPS peaks of Al(OH)₃ and AlPO₄ are the same at 74.4 eV according to the 38th and 39th references in the main text. It's necessary to check the P 2p spectrum to determine the composition of the passivation layer. Therefore, we repeated the experiment and characterization to confirm it, shown in Figure R5. The peak in Al 2p spectrum appears at 74.49 eV, very close to 74.4 eV, confirming a passivation layer outside. However, there is no apparent peak in P 2p spectrum, indicating the passivation layer being the Al(OH)₃ instead of AlPO₄. Thanks again for pointing out our mistake.

Figure R6 is the revised Figure 1 after reevaluation of the mechanisms of the HTA and

PTA.

Main text "-----"

slightly soluble lithium weak acid salts like Li_3PO_4 , Li_2CO_3 , Li_2SiO_3 , and LiAlO_2 are proposed as a new passivation additive for Al metal in an aqueous electrolyte, whose anions after hydrolysis can combine with proton, playing a role like a pH buffer in proximity to the Al foil and facilitating the formation of $\text{Al}(\text{OH})_3$.

HTAs (Li_3PO_4 , Li_2CO_3 , LiAlO_2 , Li_2SiO_3)

This indicates that Al^{3+} originating from the oxidizing of Al interact with the anion of HTAs (HPO_4^{2-} , H_2PO_4^- , HCO_3^- and HSiO_3^-) then hydrolyze to insoluble solid $\text{Al}(\text{OH})_3$

-----"

Figure R4. The distribution coefficient of phosphate species at different pH.

(Reprinted from Zhang Z, Yan L, Yu H, et al. Adsorption of phosphate from aqueous solution by vegetable biochar/layered double oxides: fast removal and mechanistic studies[J]. Bioresource technology, 2019, 284: 65-71, with permission from Elsevier)

Figure R5. The XPS spectrum of Al foil after CA experiment in 10m LiTFSI + Li_3PO_4 .

i TFSI⁻ destroys Al_2O_3 layer

ii Al corrosion without additive

ii' Al passivated by HTA

Figure R6. The HTA passivating Al mechanism schematic in an aqueous LiTFSI electrolyte. HTA: $Al(s) - e^- + H_xM^n(aq) + H_2O \rightarrow Al(OH)_3(s) + H_{x+1}M^{1-n}(aq)$. (Figure 1 in the main text).

REVIEWERS' COMMENTS

Reviewer #5 (Remarks to the Author):

The authors have well addressed the comments so I would recommend accepting the manuscript as is.